# TRF1 averts chromatin remodelling, recombination and replication dependent-break induced replication at mouse telomeres

Rosa Maria Porreca[1,2], Emilia Herrera-Moyano[1,2], Eleni Skourti[1,2], Pui Pik Law[1,2], Roser Gonzalez Franco[1,2], Alex Montoya[3], Peter Faull[3,4], Holger Kramer[3], Jean-Baptiste Vannier[1,2]*

[1]Telomere Replication and Stability group, Medical Research Council - London Institute of Medical Sciences, London, United Kingdom; [2]Institute of Clinical Sciences, Faculty of Medicine, Imperial College London, London, United Kingdom; [3]Biological Mass Spectrometry and Proteomics, Medical Research Council - London Institute of Medical Sciences, London, United Kingdom; [4]The Francis Crick Institute, Proteomics Mass Spectrometry Science and Technology Platform, London, United Kingdom

**Abstract** Telomeres are a significant challenge to DNA replication and are prone to replication stress and telomere fragility. The shelterin component TRF1 facilitates telomere replication but the molecular mechanism remains uncertain. By interrogating the proteomic composition of telomeres, we show that mouse telomeres lacking TRF1 undergo protein composition reorganisation associated with the recruitment of DNA damage response and chromatin remodellers. Surprisingly, mTRF1 suppresses the accumulation of promyelocytic leukemia (PML) protein, BRCA1 and the SMC5/6 complex at telomeres, which is associated with increased Homologous Recombination (HR) and TERRA transcription. We uncovered a previously unappreciated role for mTRF1 in the suppression of telomere recombination, dependent on SMC5 and also POLD3 dependent Break Induced Replication at telomeres. We propose that TRF1 facilitates S-phase telomeric DNA synthesis to prevent illegitimate mitotic DNA recombination and chromatin rearrangement.

*For correspondence:
j.vannier@lms.mrc.ac.uk

Competing interests: The authors declare that no competing interests exist.

## Introduction

Telomeres are specialised nucleoprotein structures at the ends of chromosomes, composed of repetitive sequences (TTAGGG repeats in mammals) (*Moyzis et al., 1988*), long non-coding RNA called TERRA and six associated proteins, TRF1, TRF2, POT1a/b, RAP1 and TIN2, that form the shelterin complex (*de Lange, 2005*). These capping structures have the crucial function of maintaining genome stability by protecting the chromosome end from being recognised as DNA double strand breaks (DSBs) (*Palm and de Lange, 2008*). They also represent challenging structures for the replication machinery, which is associated to telomere fragile sites (*Martínez et al., 2009*; *McNees et al., 2010*; *Sfeir et al., 2009*; *Vannier et al., 2012*). Telomere fragility is identified by the formation of multitelomeric signals (MTS), where telomeres appear as broken or decondensed, resembling the common fragile sites (CFS) observed at non telomeric loci after treatment with aphidicolin (APH). TRF1 facilitates the progression of the replication fork at telomeres, by recruiting a specialised DNA helicase BLM, which in turn resolve secondary structures, similar to fission yeast ortholog Taz1 (*Lee et al., 2018*; *Martínez et al., 2009*; *Miller et al., 2006*; *Sfeir et al., 2009*).

During tumorigenesis, cancer cells can achieve replicative immortality by activation of telomere maintenance mechanisms. The majority of cancer cells reactivate telomerase, while a minority (10–15%) uses an alternative mechanism named ALT for alternative lengthening of telomeres (*Bryan et al., 1997*; *Kim et al., 1994*). Intriguingly, ALT is characterised by the appearance of ALT-associated PML bodies (APBs), specialised sites where a subset of telomeres co-localises with PML protein and several DNA repair and homologous recombination (HR) proteins (*Draskovic et al., 2009*; *Wu et al., 2000*; *Yeager et al., 1999*). ALT telomeres can be maintained by more than one mechanism of recombination. Indeed, in yeast, two different ALT-like pathways have been described: type I and type II, both dependent on Rad52. While Type I requires Rad51 to mediate the invasion of a homologous sequence, Type II is Rad51 independent but Rad50 (of the MRX/N complex) dependent. Both Type I and II mechanisms require the DNA polymerase Pol32, which initiates DNA synthesis for several kilobases, in a process known as Break Induced Replication (BIR) (*Ira and Haber, 2002*). Recently, multiple groups have revisited this Rad51 independent DNA synthesis repair pathway at mammalian ALT telomeres (*Dilley et al., 2016*; *Garcia-Exposito et al., 2016*; *Roumelioti et al., 2016*). Mammalian BIR is dependent on POLD3 and POLD4, subunits of DNA polymerase delta and orthologs of yeast Pol32. ALT cells present increased DNA damage response (DDR) and several studies have underlined the contribution of replication stress to ALT-mediated telomere extension (*Arora et al., 2014*; *Cox et al., 2016*; *Pan et al., 2017*). However, the molecular mechanisms initiating recombination in ALT cells are still unclear.

In order to gain insight into the chromatin composition of telomeres undergoing replication stress, we performed Proteomics of Isolated Chromatin segments (PICh), using *TRF1* conditional knock-out Mouse Embryonic Fibroblasts (MEFs, telomerase positive). Surprisingly, we found that telomeres lacking TRF1 are enriched in SMC5/6, DNA polymerase δ (POLD3), and chromatin remodelling factors known to be associated with ALT telomeres. These cells also present additional DNA damage and recombination hallmarks such as formation of APBs, mitotic DNA synthesis at telomeres, a feature of BIR and increased TERRA levels. Further investigation using specific shRNAs against the SMC5/6 complex or POLD3 revealed how these two complexes are key regulators of the recombination signature identified in *TRF1* deleted cells. Taken together, these results strongly identify TRF1 as a central player in preserving telomeric chromatin against HR, induced by DNA replication stress, and particularly POLD3 dependent-mitotic DNA synthesis.

## Results

### Capture of TRF1 depleted telomeres by PICh reveals drastic changes in the chromatin composition

To isolate and identify the chromatin composition of TRF1 depleted telomeres, we employed Proteomics of Isolated Chromatin segments (PICh), a powerful and unbiased technique that uses a desthiobiotinylated oligonucleotide complementary to telomeric repeat sequences to specifically pull down telomeric chromatin (*Déjardin and Kingston, 2009*). We performed PICh in MEFs harboring a *TRF1* conditional allele. MEFs lacking *TRF1* are well known to undergo replicative stress; however, they can grow for up to 8 days before entering senescence, making them optimal for investigating replication stress at telomeres (*Martínez et al., 2009*; *Sfeir et al., 2009*). Cells were transduced twice (day 0 and 3) with a CRE or GFP control adenovirus and collected 7 days after the first transduction, as indicated in the timeline (*Figure 1A*). Excision of exon 1 of *TRF1* by CRE recombinase (*Sfeir et al., 2009*) resulted in the expected loss of TRF1 protein as determined by immunoblotting and deficient cells do not show drastic change in the cell cycle distribution (*Figure 1—figure supplement 1*). Cells were fixed and isolation of telomeres was performed using a probe complementary to TTAGGG repeats or a scrambled probe as a negative control. Finally, telomeric chromatin was isolated from both control cells (wt) and *TRF1* deleted cells before mass spectrometry identification (*Figure 1C*). We identified a list of 1306 proteins that was subjected to refinement in order to remove unspecific bound proteins or contaminants found with the scrambled probe (see experimental procedure for detailed description). Based on the analysis of label free quantification (LFQ intensities), we found 119 proteins presenting a gain of abundance at TRF1 depleted telomeres (Log2<-2) and 206 factors were displaced from these telomeres (Log2>-2), considering that a cut-off for differential expression is set to log2 fold change (TRF1deletion/wt)> |2| and -Log (p-value) > 1

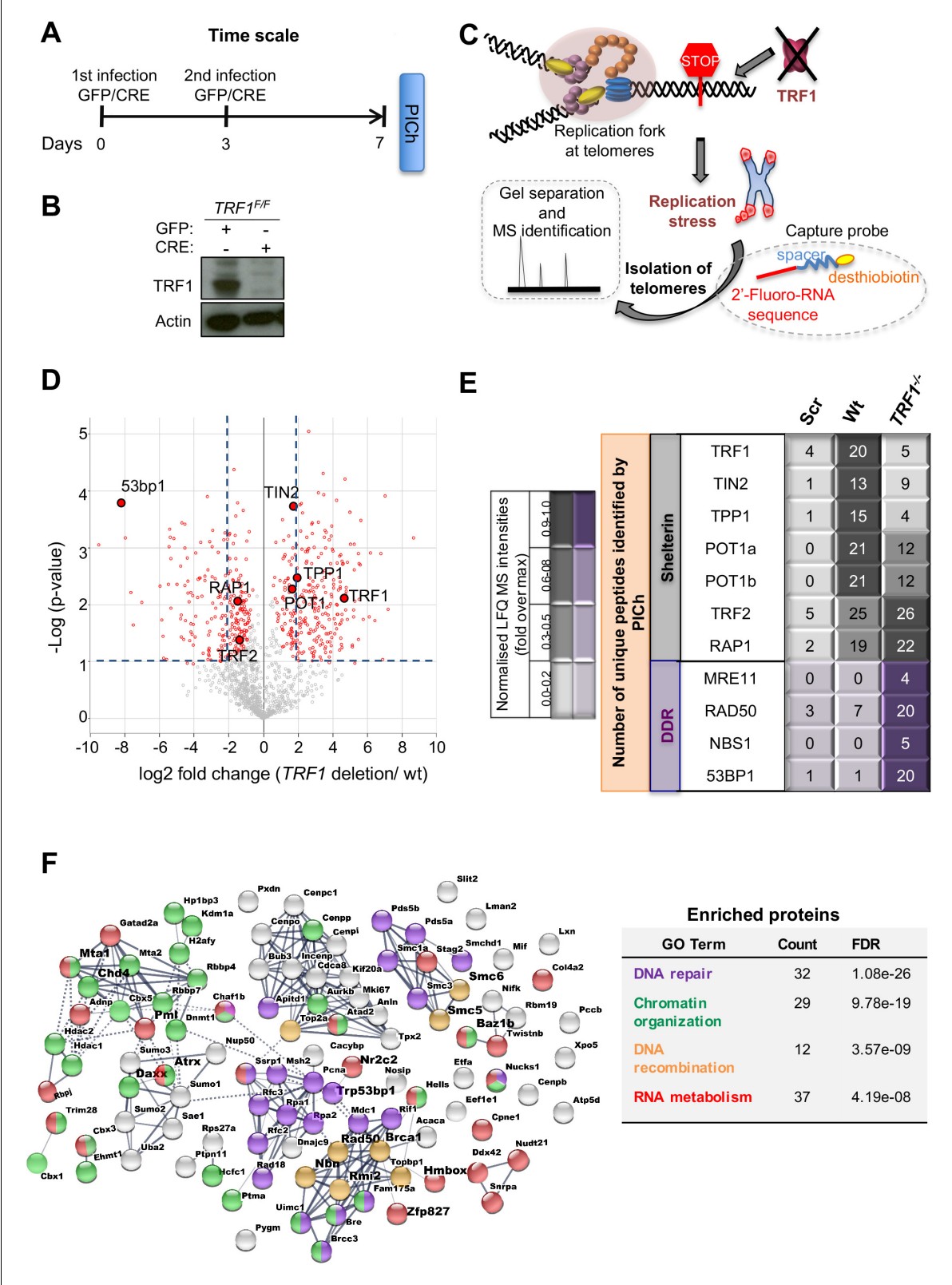

**Figure 1.** Proteomics of isolated chromatin segments (PICh) of TRF1 depleted mouse telomeres. (**A**) Overview of experimental timeline aimed at performing PICh experiment after induction of *TRF1* deletion. *TRF1*^F/F^ MEFs were infected twice (day 0 and 3) with adenovirus containing either GFP-control or CRE and collected at day 7 for PICH experiments. (**B**) Western blot showing deletion of *TRF1* in MEFs after infections with CRE Adenovirus, at day 7 as in A. (**C**) Schematic representation of the PICh analysis performed to detect chromatin changes occurring at telomeres upon *TRF1* deletion.

*Figure 1 continued on next page*

*Figure 1 continued*

(D) Volcano Plot based on LFQ intensities of proteins. Cut off for differential expression were set to log2 fold change (TRF1deletion/wt)> |2| and -Log (p-value) > 1. (E) Table listing shelterin components and some of the DNA damage response (DDR) factors identified. The corresponding number of unique peptide isolated is indicated for each factor of interest. Relative LFQ intensity abundance profiles were visualised in the form of a heat-map, by scaling each protein intensity to the maximum intensity across conditions. Light to darker colours indicate increasing relative protein abundance. (F) Connectivity map for proteins recruited at telomeres upon TRF1 deletion using string-db.org software. Solid lines, represents strong direct interactions, while dashed lines represent no evidence for direct interaction. In violet, DNA damage and repair proteins; in orange, factors belonging to DNA repair specifically involved in DNA recombination process; while in green and red, important factors for chromosome maintenance and factors involved in RNA metabolism, respectively. Source data are provided as a Source Data File.

The online version of this article includes the following source data and figure supplement(s) for figure 1:

**Source data 1.** *TRF1^{F/F}* MEFs western blot.
**Source data 2.** PICh data (*TRF1^{F/F}* telo vs *TRF1^{-/-}* telo).
**Source data 3.** PICh data.
**Figure supplement 1.** *TRF1* deficient MEFs show no drastic changes in their cell cycle.

(*Figure 1D*). Amongst these 206 proteins, we found TRF1, as expected due to the knock-out of its gene, but also one component of the CST complex (CTC1), important player in the efficient restart of stalled replication forks at telomeres (*Gu et al., 2012*) and recruited through POT1b interaction (*Wu et al., 2012*). Interestingly, POT1b is also less abundant at TRF1 depleted telomeres (*Figure 1D–E*). On the other end, the group of 119 proteins enriched in *TRF1* deleted cells includes several factors involved in structural maintenance of chromosomes (SMC), HR and DNA damage response (*Figure 1D–E–F*), such as the MRN complex (MRE11, RAD50 and NBS1). The identification of 53BP1 recruited to TRF1 depleted telomeres (*Figure 1D–F*) acts as a positive marker for the specificity of this proteomic analysis, as reported before in *Martínez et al. (2009)*; *Sfeir et al. (2009)*. Moreover, we could identify drastic and previously uncharacterised changes of the telomeric proteome at telomeres undergoing replication stress presented hereafter.

## TRF1 suppresses APBs formation and HR at telomeres

Interestingly, *TRF1* deficient MEFs present a telomeric enrichment for factors involved in HR and chromatin remodelling (BRCA1, PML, SMC5/6, ATRX and, NurD associated factors) that are usually abundant at ALT telomeres (*Figure 2A*) (*Conomos et al., 2014*; *Draskovic et al., 2009*; *Marzec et al., 2015*; *Potts and Yu, 2007*). To validate the specific association of some of these factors with TRF1 depleted telomeres in telomerase positive MEFs, we carried out chromatin immunoprecipitation (ChIP) experiments using ChIP-grade specific antibodies followed by telomeric dot-blot. TRF1 antibody was used as a negative control for our experiment, while the recruitment of BAZ1b, BRCA1 and some subunits of the nucleosome remodelling and deacetylase (NurD) complex (p66a, MTA1, ZNF827 and CHD4) was assessed. For all these factors, with the exception of p66a for which no statistical significance was achieved, we observed a specific enrichment at telomeres upon *TRF1* deletion (*Figure 2—figure supplement 1, A*). In addition, to confirm the presence of PML at telomeres lacking TRF1, as suggested by our PICh data (*Figure 2A*), we performed immuno-FISH and scored for the formation of APBs. We observed a two-fold increase in the number of co-localisations between PML and telomeres in *TRF1^{-/-}* MEFs compared to control cells (*Figure 2C*). To test if the recruitment of some of these factors is a consequence of chronic replication stress or if TRF1 directly inhibit these events, we treated wt MEFs with low doses of APH (0.4 µM) for 3 days. Using ChIP-dot blot, we did not detect the recruitment of any of the chromatin remodellers and DNA repair factors BRCA1, MTA1, CHD4, ZNF827 and BAZ1b, present at TRF1 depleted telomeres (*Figure 2—figure supplement 1, B*). Furthermore, we show that wt MEFs treated with APH do not form APBs despite a general increase in PML foci (*Figure 2—figure supplement 1, C*). Overall these data demonstrate that telomeres depleted of TRF1 engage chromatin remodelling associated with the formation of APBs, platform of recombination (*Cesare and Reddel, 2010*). Hence, to test if TRF1 is a suppressor of recombination, we revisited the incidence of telomeric sister chromatid exchanges (T-SCE) using chromosome orientation FISH (CO-FISH) in *TRF1* deficient cells (*Figure 2D*). We identified an increase in T-SCE in *TRF1^{-/-}* MEFs (2.8%) compared to control cells (0.4%) (*Figure 2D*). This result is at odds with previous publications where T-SCE events detected at TRF1 depleted telomeres were not significantly enriched, with only 1% of T-SCEs detected compared to 0.1% in wt cells

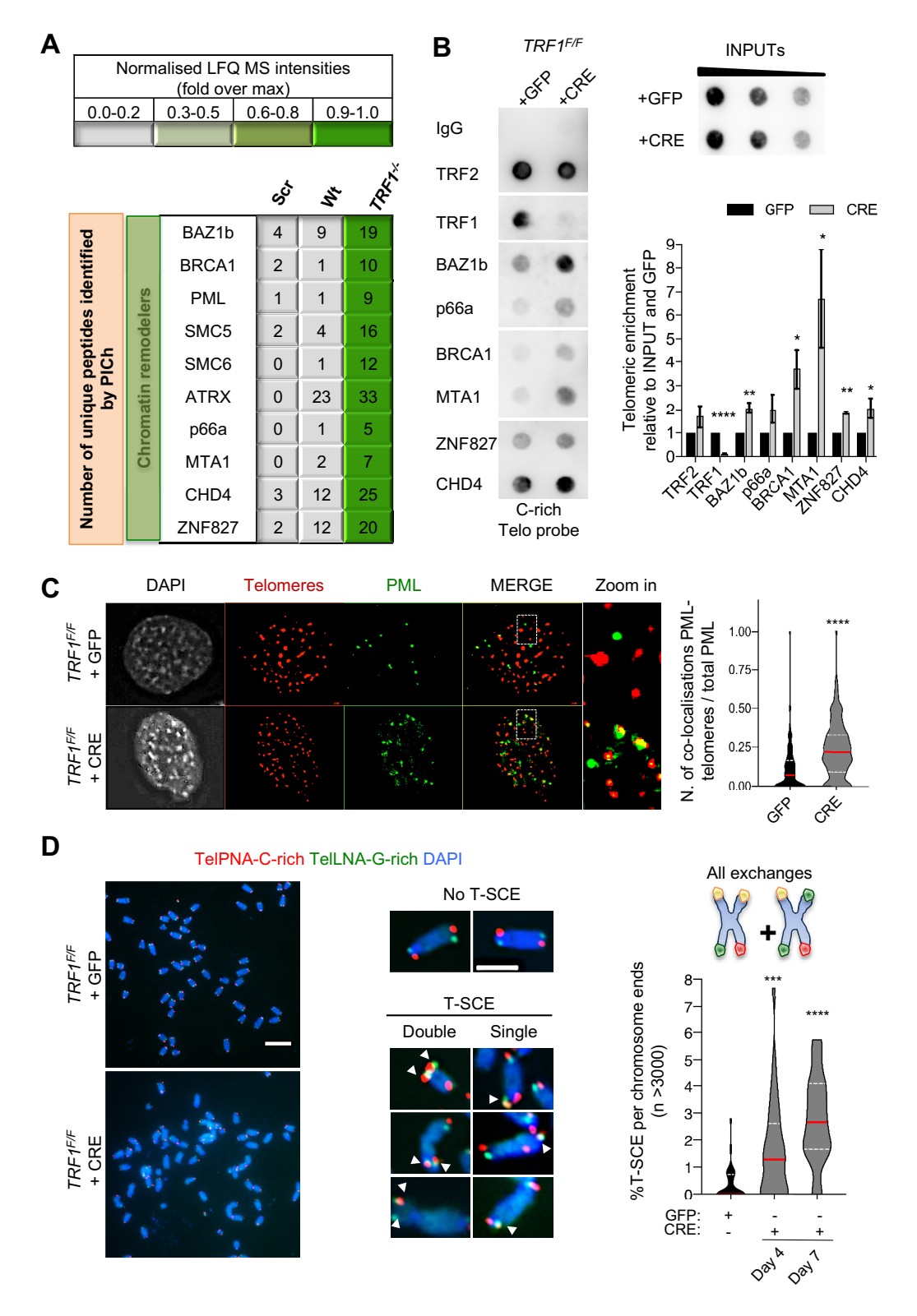

**Figure 2.** Recombination factors are recruited at TRF1 depleted telomeres. (A) Table listing chromatin remodellers identified. The corresponding number of unique peptide isolated is indicated for each factor of interest. Same as in *Figure 1*, light to darker colours indicate increasing relative protein abundance. (B) Validation of chromatin remodeller factors by ChIP-dot blot analysis in wt (+GFP) and *TRF1⁻/⁻* (+CRE) conditions using ChIP grade antibodies against chosen factors after chromatin preparation from MEFs. The blot was revealed with a DIG-Tel-C-rich probe. ChIP signals were

*Figure 2 continued on next page*

*Figure 2 continued*

normalised to DNA input and GFP control. Data are represented as telomeric enrichment of proteins relative to GFP ± SEM of at least three independent biological replicates. P values, two-tailed student t-test (*, p<0.05; **, p<0.01; ****, p<0.0001). (C) Representative image of Immunofluorescence showing co-localisation of Telomeres (red) with PML (green) in MEFs nuclei (DAPI) treated with GFP and CRE. Data from two independent biological replicates are represented as number of Telomeres-PML co-localising foci divided by the total number of PML present per nucleus (n = 300 nuclei) with a violin plot, where the median is underlined in red and quartiles in white. P values, two-tailed student t-test (****, p<0.0001). Source data are provided as a Source Data File. (D) Representative images of the chromosome-oriented CO-FISH assay with denaturation, used to score for telomeric T-SCEs in *TRF1^{F/F}* MEFs infected with GFP or CRE. Telomeres are labelled with TelPNA-C-rich-Cy3 (red) and TelLNA-G-rich-FAM (green), while chromosomes are counterstained with DAPI (blue). Scale bar, 10 μm. Enlarged intersections show the difference between a chromosome with No T-SCE (top) and a chromosome with T-SCE (bottom). T-SCE images show double T-SCEs (left) and single chromatid events (right). Scale bar, 2 μm. For quantification, T-SCE was considered positive when involved in a reciprocal exchange of telomere signal with its sister chromatid (both telomeres yellow) and for asymmetrical exchanges at single chromatid (one telomere yellow). Data are indicated as % of T-SCE per sister telomere (n = > 3000 chromsome ends) and are represented with a violin plot, where the median is underlined in red and quartiles in white. P value, two-tailed student t-test (****, p<0.0001).

The online version of this article includes the following source data and figure supplement(s) for figure 2:

**Source data 1.** Control specificity of proteomic binding (*TRF1^{-/-}* telo vs *TRF1^{-/-}* scbl).
**Source data 2.** ChIP quantification in *TRF1^{F/F}* MEFs.
**Source data 3.** Quantification of APBs in *TRF1^{F/F}* MEFs.
**Source data 4.** T-SCEs quantification in *TRF1^{F/F}* MEFs.
**Figure supplement 1.** Validation of telomeric ChIP with ALU probe and effect of APH on telomeric chromatin and APBs.
**Figure supplement 1—source data 1.** Alu probe control for ChIP quantification in *TRF1^{F/F}* MEFs.
**Figure supplement 1—source data 2.** ChIP quantification in MEFs treated with APH.
**Figure supplement 1—source data 3.** Quantification of APBs in MEFs treated with APH.
**Figure supplement 2.** TRF1 loss and APH replication stress induce different types of telomeric recombination.
**Figure supplement 2—source data 1.** T-SCEs quantification in *TRF1^{F/F}* MEFs.
**Figure supplement 2—source data 2.** T-SCEs quantification of MEFs treated with APH.

(*Martínez et al., 2009*; *Sfeir et al., 2009*). In fact, this discrepancy might be explained by the difference in timing for the analysis of T-SCEs in *TRF1* deficient cells. Both publications report the lack of recombination effect by T-SCEs at 3 or 4 days after *TRF1* loss, while we generally carry our investigations at day 7. Therefore, we repeated the experiments in *TRF1^{-/-}* cells at different time points post infection: day 4 and day 7, finding respectively 1.6% and 2.8% of T-SCEs per chromosome end (*Figure 2—figure supplement 2, A*-left graph), indicating a lower % of T-SCE events happening at earlier time point. A second distinct difference with previous reports is the type of telomere signal exchanges that we analysed. As in *Sfeir et al. (2009)*, all types of telomere signal exchanges (e.g. the exchanges appearing at single chromatids and the reciprocal exchanges at both chromatids) were considered. However, *Martínez et al. (2009)* only refers to reciprocal exchanges at both chromatids. Thus, we next classified T-SCEs detected in *TRF1* deficient MEFs into these two different types (single and double) and found that 4 days post infection only T-SCEs at single chromatids were significantly increased (*Figure 2—figure supplement 2, A*-right graph), while the reciprocal exchanges were not enhanced at TRF1 depleted telomeres (*Figure 2—figure supplement 2, A*-middle graph). Therefore, our detailed analysis of the nature and timing of T-SCEs in *TRF1* deficient MEFs not only demonstrates an unappreciated role of TRF1 in suppressing HR, but also suggests that single chromatid exchanges, characteristic of conservative DNA synthesis – Break Induced Replication (*Roumelioti et al., 2016*), are the initial events arising in *TRF1^{-/-}* MEFs. To further discriminate between suppression of recombination by TRF1 and replication-dependent recombination, we quantified T-SCEs in mouse cells treated with APH. Like for TRF1-depleted telomeres, we also detected an increase in single exchanges in APH-treated MEFs (*Figure 2—figure supplement 2*, B-right panel). The absence of reciprocal exchanges between telomere chromatids indicates that APH-dependent replication stress does not trigger an HR repair but only BIR, albeit we cannot completely exclude that longer replication stress could lead to HR (*Figure 2—figure supplement 2*, B-left panel). Overall our data highlights the specific function of TRF1 in suppressing chromatin remodeling, APBs and HR.

## TRF1 depletion causes TERRAs upregulation in mouse cells

Since depleting telomeres of TRF1 induces chromatin remodelling, formation of APBs and increase of HR, we next decided to revisit the effect of TRF1 on TERRA molecules, which are proposed to regulate telomere recombination (*Yu et al., 2014*). Previous studies have reported in-vivo interactions between TRF1 and TERRA (*Deng et al., 2009*) and also a possible transcriptional regulation by TRF1 through a mechanism involving RNA polymerase II - TRF1 interaction (*Schoeftner and Blasco, 2008*). However, the role of TRF1 regulating telomere transcription appears complex since contrasting results have been reported by different groups in both human and mouse cell lines (*Lee et al., 2018*; *Schoeftner and Blasco, 2008*; *Sfeir et al., 2009*). We performed both RNA dot-blot and Northern-blot analyses showing a significant increase in TERRA molecules upon loss of *TRF1* in immortalised MEFs, 7 days after transduction (*Figure 3A–B*) but also at earlier time point (day 4) and in primary MEFs (*Figure 3—figure supplements 1, A–B–C*). Collectively, we identify an increase in TERRA molecules upon TRF1 removal from telomeres, which suggests a TRF1 function in suppressing mouse TERRAs. Particularly, we noticed that these TERRAs have high molecular weight and can only be detected when an alkaline treatment is performed during Northern-blotting (*Figure 3—figure supplement 1, D*). In addition, we carried out TERRA-FISH (*Figure 3C*) in MEFs deficient for *TRF1*, confirming a significant increase in numbers and intensity of TERRA foci per nucleus (*Figure 3C*). To determine whether this phenotype is a direct consequence of replication stress, MEFs were treated with low doses of APH and TERRA analysis was carried out by RNA dot-blot and Northern blot. Similar to TRF1 depleted cells, APH-treated MEFs exhibited a significant increase in TERRAs (*Figure 3—figure supplement 1, E–F*). Taken together, these results suggest that TRF1-dependent replication stress enhances telomere transcription or TERRAs stability.

## mTRF1-dependent APBs formation and BIR are conserved in human cells

The presence of APBs, BIR, HR and TERRAs, in *TRF1⁻/⁻* MEFs indicates some similarities with ALT telomeres. However, the absence of telomere heterogeneity, c-circle formation and still presence of telomerase activity (*Figure 3—figure supplements 2, A–B–C*) also suggest that this ALT-like phenotype is not complete. To test whether the function of TRF1 in suppressing these ALT-like features is conserved in human cells, we used small RNA interference against TRF1 in HT1080-ST cells (long telomeres), performing all the analysis 6 days post-transfection (*Figure 4A*). TRF1-depleted HT1080-ST cells present a modest but significant increase in APBs formation (*Figure 4B*). This phenotype appears to be milder as compared to *TRF1⁻/⁻* MEFs, probably because HT1080/ST present higher levels of APBs compared to other human (*Pickett et al., 2009*) or mouse cells (*Figure 2C*). We conclude that TRF1 function is somewhat maintained between the two species. To assess the human TRF1 function in suppressing HR and BIR, we measured sister chromatid exchanges at telomeres in HT1080-ST depleted for TRF1 (*Figure 4C*). We observed a significant increase in single exchanges (*Figure 4C*-right panel) however, we could not detect double exchanges in cells transfected with TRF1 siRNA compared to the ctl (*Figure 4C*-left panel). Finally, TERRA expression was analysed by DNA dot blot and no differences were observed between ctr and TRF1 depleted cells (*Figure 4—figure supplement 1, A*). To evaluate whether long term depletion of TRF1 in human cells would cause any increase in TERRA expression, as observed in *TRF1⁻/⁻* MEFs, we used a doxycycline inducible TRF1 CRISPR/Cas9 HeLa cell line (*Figure 4—figure supplement 1, B*) (*Kim et al., 2017*). However, even 15 days after hTRF1 deletion, no significant increase in TERRAs was detected. Human TRF1 KO cells did not show any growing defects after 15 days in culture, which is remarkably different from TRF1 deficient MEFs that fail to proliferate due to a rapid induction of senescence (*Karlseder et al., 2003*; *Martínez et al., 2009*).

In conclusion, some of TRF1 functions are conserved between mouse and human. For both species, TRF1 suppresses telomere fragility and therefore is essential to facilitate DNA replication through telomeric repeats (*Lee et al., 2018*; *Sfeir et al., 2009*). We now show that like for mouse TRF1, its human homolog is essential to suppress APBs formation and BIR, which is likely responsible for the rescue of stalled/collapsed telomeric forks.

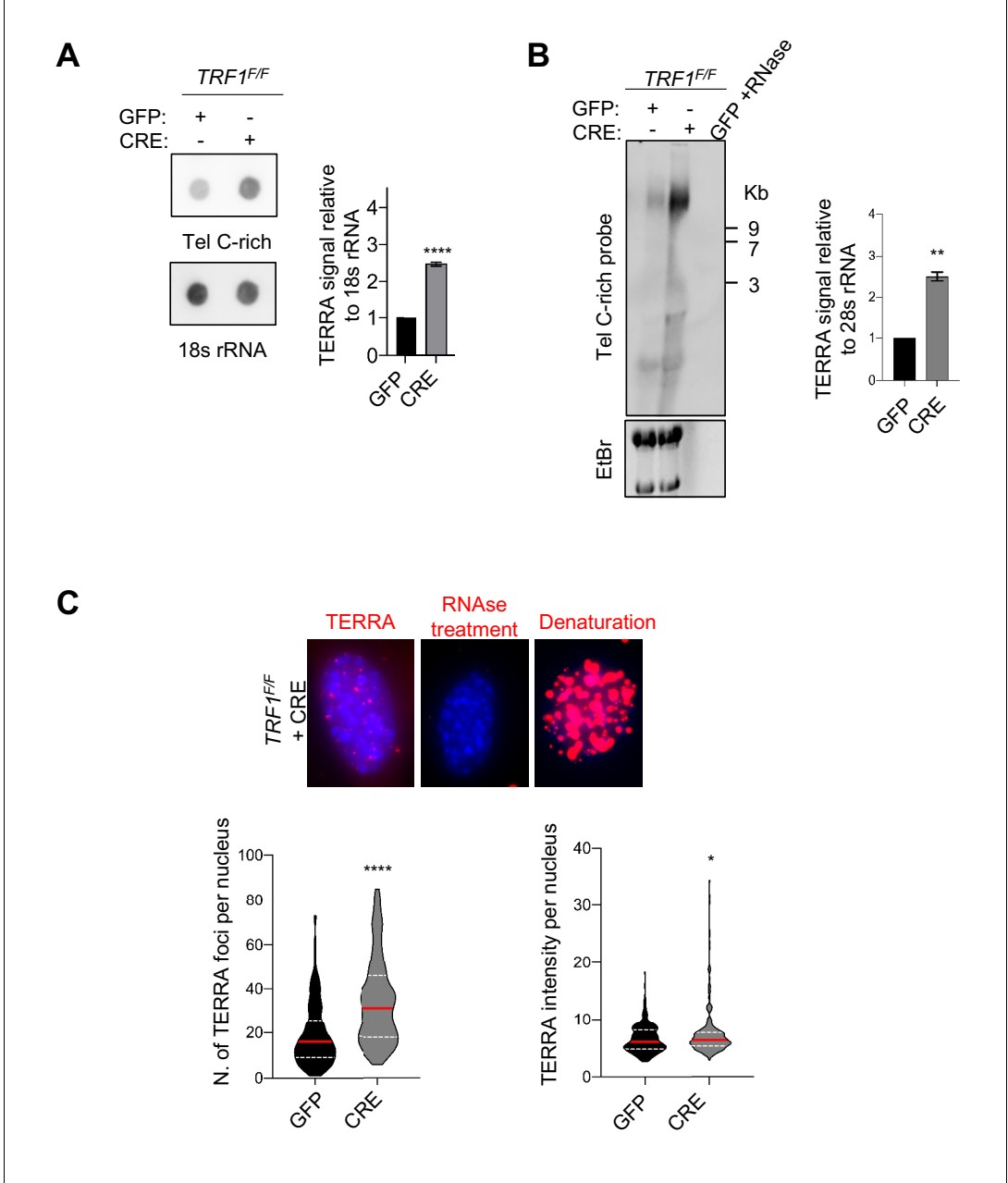

**Figure 3.** TRF1 depletion causes TERRAs upregulation. (A) RNA dot blot analysis in wt and *TRF1* deleted MEFs. The blot was revealed with a DIG-Tel-C-rich probe or 18 s rRNA as a control. TERRA signals were normalised to 18 s rRNA and GFP control ± SEM of at least three independent biological replicates. P values, two-tailed student t-test (****, p<0.0001). (B) TERRA detection by Northern blotting upon *TRF1* deletion. The blot was revealed with a DIG-Tel-C-rich probe (upper part). Ethidium bromide (EtBr) staining (bottom) of rRNAs was used as loading control. TERRA signals were normalized to 28 s rRNA signal from EtBr staining ± SEM of 2 independent biological replicates. P values, two-tailed student t-test (**, p<0.01). (C) Representative images of TERRA-FISH experiment (top panel) showing the difference between cells stained with TERRA (red), negative control with RNAse A treatment and positive control after denaturation. TERRA-FISH quantification (bottom panel) in wt (+GFP) and *TRF1⁻/⁻* (+CRE) conditions. Violin plots are representing the number of TERRA foci (left) and TERRA intensity (right) (n = 250) per nucleus, where the median is underlined in red and quartiles in white, two-tailed student t-test (****, p<0.0001); Mann-Whitney test used for TERRA intensity quantification (*, p<0.05).

The online version of this article includes the following source data and figure supplement(s) for figure 3:

**Source data 1.** Quantification of Telomeric RNA molecules by dot-blot in *TRF1^F/F^* MEFs.

**Source data 2.** Quantification of TERRAs by Northern.

**Source data 3.** Quantification of number of TERRA foci.

**Figure supplement 1.** TRF1 deletion and APH replication stress cause increased TERRA levels in immortalised MEFs (day4) and also in primary MEFs (day 6).

*Figure 3 continued on next page*

*Figure 3 continued*

**Figure supplement 1—source data 1.** WB of *TRF1^{F/F}* primary MEFs and immortalized after 4 days of CRE.

**Figure supplement 1—source data 2.** Telomeric RNA molecules by dot-blot in primary *TRF1^{F/F}* MEFs and immortalized after 4 days of CRE.

**Figure supplement 1—source data 3.** Quantification of Telomeric RNA molecules by dot-blot in primary *TRF1^{F/F}* MEFs and immortalized after 4 days of CRE.

**Figure supplement 1—source data 4.** TERRAs by Northern in *TRF1^{F/F}* MEFs.

**Figure supplement 1—source data 5.** Quantification of Telomeric RNA molecules by dot-blot in wt MEFs treated with APH.

**Figure supplement 1—source data 6.** TERRAs by Northern in wt MEFs treated with APH.

**Figure supplement 2.** *TRF1* deficient MEFs present normal telomere distribution, telomerase activity and no c-circles.

**Figure supplement 2—source data 1.** Telomere length by Southern in *TRF1^{F/F}* MEFs.

**Figure supplement 2—source data 2.** Telomerase activity by TRAP in *TRF1^{F/F}* MEFs.

**Figure supplement 2—source data 3.** c-circle amplification assay in *TRF1^{F/F}* MEFs and U2OS (+ctl).

## TRF1 suppresses mitotic DNA synthesis at telomeres

Since the denaturing CO-FISH experiments in *TRF1* deficient cells identified single chromatid exchanges that are proposed to be reminiscent of BIR events, an HR alternative pathway required in G2-M phase (*Roumelioti et al., 2016*), we tested whether TRF1 depleted telomeres trigger non S-phase DNA synthesis. We performed a pulse with 5-bromo-2-deoxyuridine (BrdU) for 2 hr (*Figure 5A*) before carrying out BrdU immunofluorescence at telomeres in interphase cells (*Figure 5B–C*). Only non S-phase cells were counted in this experiment, based on the formation of clear BrdU foci (*Dilley et al., 2016*; *Nakamura et al., 1986*) (*Figure 5B*). *TRF1^{-/-}* MEFs display elevated BrdU incorporation at telomeres, showing two-times more telomere synthesis compared to control cells (*Figure 5C*). To investigate DNA synthesis happening exclusively in mitosis, so-called MiDAS (*Minocherhomji et al., 2015*), we performed a similar experiment in metaphases. After incubating wt and *TRF1* deficient MEFs with 5-ethynyl-2-deoxyuridine (EdU) and colcemid for 1 hr, mitotic cells were collected to analyse EdU incorporation on metaphase chromosomes (*Figure 5A*). We scored for telomeric and non-telomeric EdU foci (mitotic DNA synthesis) and found that CRE induced cells had a significant increase in telomeric mitotic DNA synthesis compared to the GFP control cells (*Figure 5D*). This result confirms that TRF1 depleted telomeres present an increased level of non-S-phase DNA synthesis, similar to what is observed in ALT cells. In addition, analysis of EdU incorporation in metaphase spreads allowed us to distinguish between conservative BIR associated DNA synthesis and HR semi-conservative DNA synthesis (*Min et al., 2017*). In the first case, EdU would be labelled on a single chromatid (*Figure 5E* upper panel), while in the latter, EdU would localise to both chromatids (*Figure 5E* bottom panel). Thus, to assess the mechanism of DNA synthesis in *TRF1* deleted cells, the pattern of EdU incorporation on metaphase chromosomes was further investigated (*Figure 5F*). Non-telomeric (upper panel) and telomeric (middle panel) EdU foci formed mainly on a single chromatid. In fact, 72% of the mitotic DNA synthesis at non-telomeric sites localised to a single chromatid, while the remaining 28% of the signal was present at both chromatids (*Figure 5F*, upper panel). This result is even more striking when EdU signal was restricted to telomeres, with almost all the co-localisation being present at single chromatids (95%). These observations suggest that TRF1 is crucial for the suppression of mitotic DNA synthesis mediated by BIR at telomeres.

## Mitotic DNA synthesis at replication stressed telomeres is POLD3 dependent

BIR is a recombination dependent process reinitiating DNA replication when one end of a chromosome shares homology with the template DNA, leading to conservative DNA synthesis, which is dependent on RAD52 and POLD3 (*pol32* homolog in yeast) (*Bhowmick et al., 2016*; *Sotiriou et al., 2016*). ALT telomeres have recently been reported to be elongated by BIR, in a POLD3 and SMC5-dependent manner (*Dilley et al., 2016*; *Min et al., 2017*; *Potts et al., 2006*). Since the SMC5/6 complex was exclusively enriched in PICh purified TRF1 depleted telomeres (*Figure 2A*), we further investigated the role of POLD3 and SMC5 in BIR DNA synthesis observed in *TRF1^{-/-}* MEFs. We generated *TRF1^{F/F}* cells deficient in *SMC5* or *POLD3* using specific shRNAs. Upon infection with GFP or CRE adenovirus, we produced respectively single or double deletion *TRF1-SMC5* or *TRF1-POLD3*

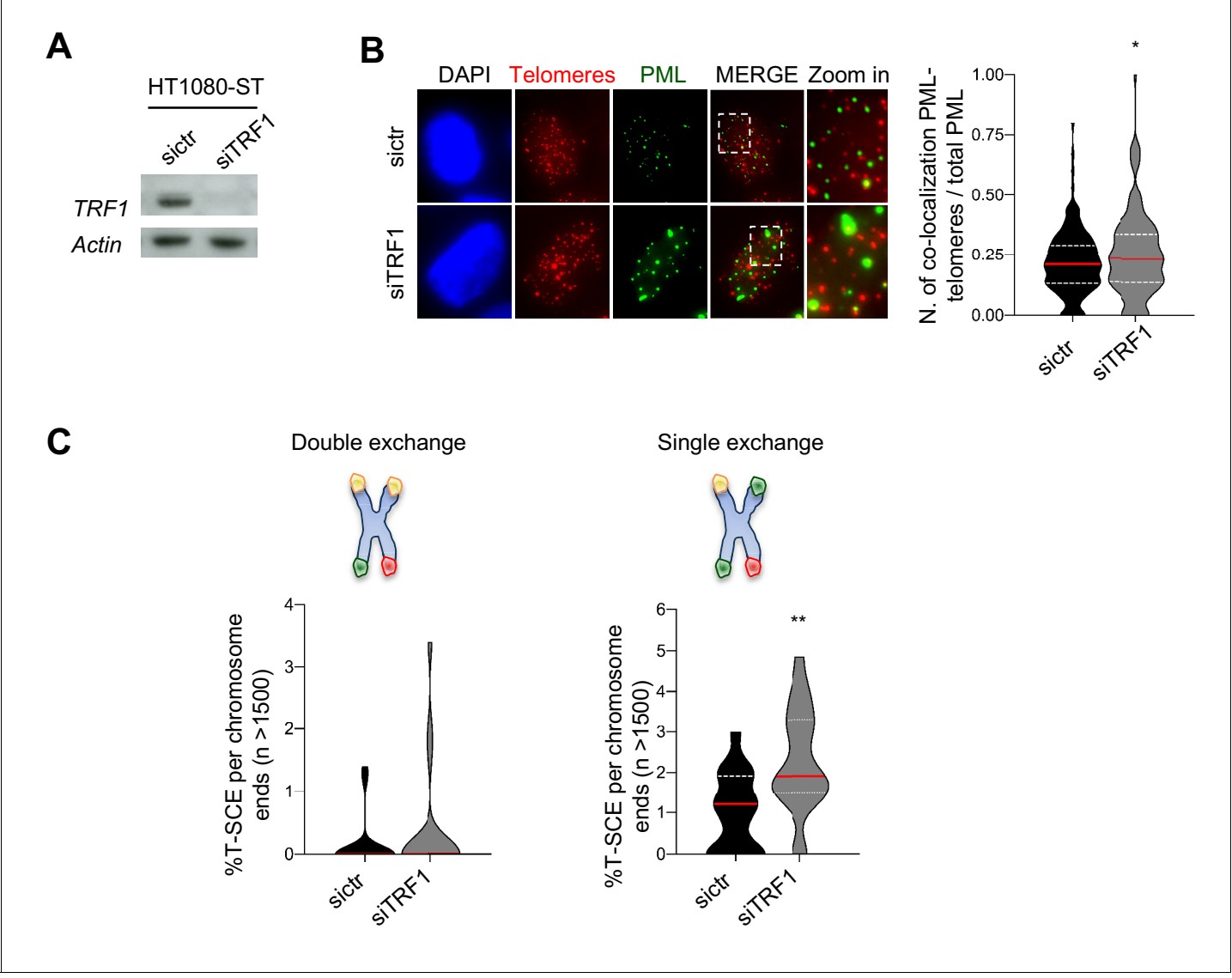

**Figure 4.** Human TRF1 suppresses APBs formation and BIR. (**A**) Western blotting showing expression of TRF1 and Actin (loading control) proteins in human HT1080-ST cell line after depletion of TRF1 with siRNAs (6 days post transfection). sictr is used as negative control for transfection. (**B**) Representative image of Immunofluorescence showing co-localisation of Telomeres (red) with PML (green) in HT1080-ST nuclei (DAPI) transfected with sictr or siTRF1. Data are represented as number of Telomeres-PML co-localising foci divided by the total number of PML present per nucleus (n = 150 nuclei) and are shown as a violin plot, where the median is underlined in red and quartiles in white. P values, two-tailed student t-test (*, p<0.05). Source data are provided as a Source Data File. (**C**) Quantification of telomeric T-SCEs in HT1080-ST transfected with sictr or siTRF1. Telomeric exchanges are classified as double exchanges (reciprocal, both chromatids-left graph) and single exchanges (asymmetrical, single chromatid-right graph). Violin plots represent % of T-SCE per chromosome ends (n = at least 1500 events were scored), where the median is underlined in red and quartiles in white. P value, two-tailed student t-test (**, p<0.01).

The online version of this article includes the following source data and figure supplement(s) for figure 4:

**Source data 1.** KD of TRF1 in HT1080-ST cells.
**Source data 2.** Quantification of APBs in HT1080-ST TRF1 KD.
**Source data 3.** T-SCEs quantification in HT1080-ST TRF1 KD.
**Figure supplement 1.** Human TRF1 does not influence TERRAs.
**Figure supplement 1—source data 1.** Quantification of Telomeric RNA molecules by dot-blot in HT1080-ST TRF1 KD.
**Figure supplement 1—source data 2.** KO efficiency and quantification of Telomeric RNA molecules by dot-blot in Dox inducible HeLa CRISPR/Cas9 system.

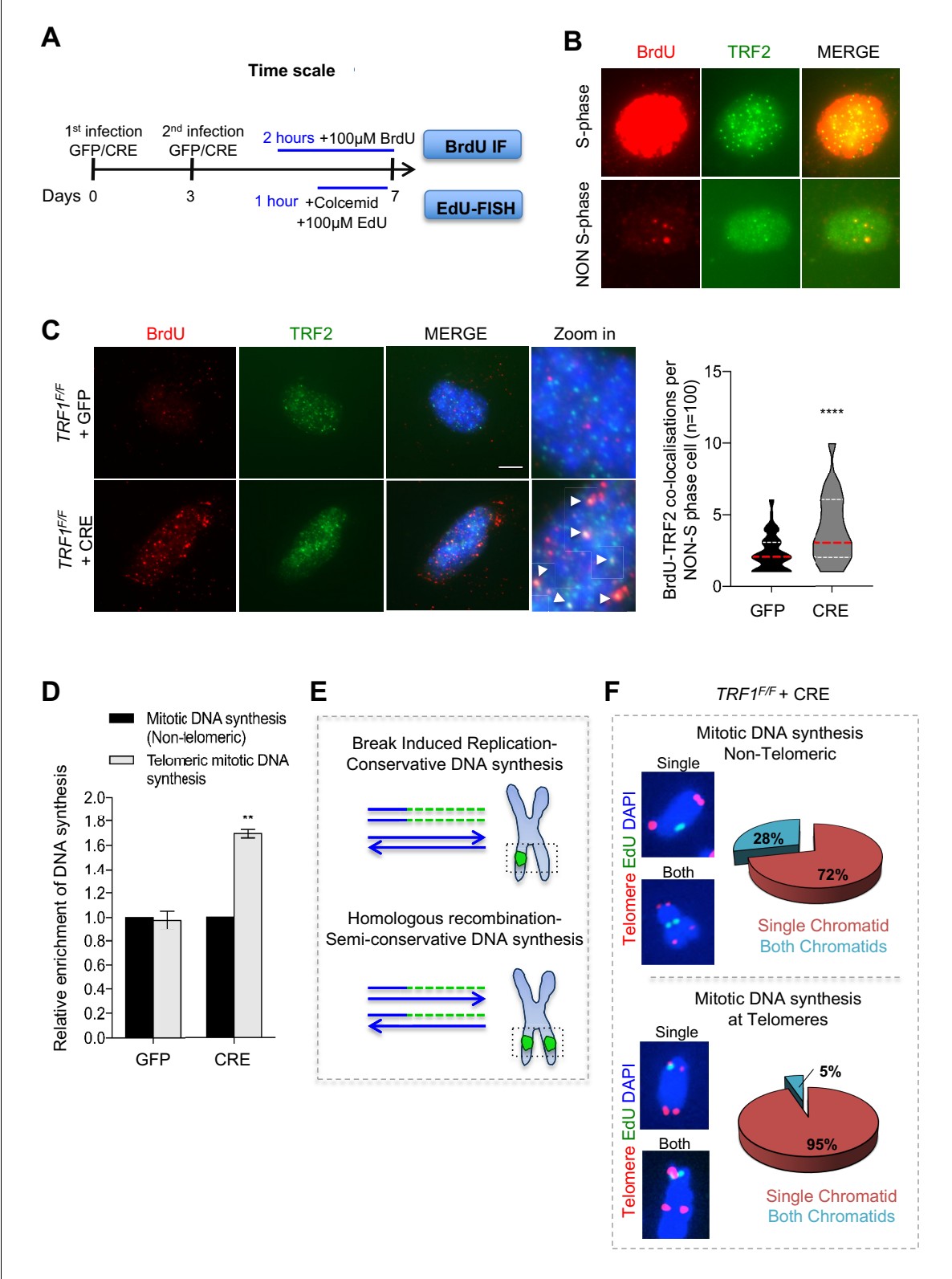

**Figure 5.** Deletion of *TRF1* induces mitotic DNA synthesis at telomeres. (**A**) Schematic overview of the experimental timeline. *TRF1^F/F^* MEFs cells were infected twice (day 0 and 3) with adenovirus containing either GFP control or CRE to mediate *TRF1* deletion. Prior to collection at day 7, cells were treated with either BrdU (100 μM) for 2 hr or EdU (100 μM) + colcemid for 1 hr, to perform respectively BrdU-Immunofluorescence (IF) or EdU-FISH on metaphases. (**B**) Representative image of BrdU (red) - TRF2 (green) immunofluorescence showing example of cells in S-phase (upper panel) and non-S-

*Figure 5 continued on next page*

*Figure 5 continued*

phase (bottom panel). (**C**) Immunofluorescence showing co-localisation of BrdU (red) with TRF2 (green) in *TRF1^{F/F}* MEFs nuclei (DAPI, blue) treated with GFP and CRE. Scale bar, 5 μm (denaturing conditions). Data are represented in a violin plot as % of cells in non-S-phase showing BrdU-TRF2 co-localising foci (n = 100 nuclei), where the median is underlined in red and quartiles in white, two-tailed student t-test (****, p<0.0001). (**D**) Quantification of DNA synthesis using the number of EdU-positive intra-chromosomes or telomeres in *TRF1^{F/F}* cells infected with GFP and CRE relative to the GFP control (n = 50 metaphases). Data are represented as relative enrichment to the GFP control ± SEM of 3 independent biological replicates. P values, two-tailed student t-test (**, p<0.01). (**E**) Schematic representation of Break Induced Replication (top part) with single EdU foci at a single chromatid and Homologous recombination (bottom part) with EdU foci at both chromatids. (**F**) Analysis of DNA synthesis in *TRF1* deleted cells. *Upper panel:* Non-telomeric mitotic DNA synthesis. Representative images showing EdU signal (green) in a single chromatid or in both chromatids. Pie chart representing % of chromosomes having EdU signal at a single chromatid or at both chromatids. *Bottom panel:* Telomeric mitotic DNA synthesis. Representative images showing EdU signal (green) at telomeres (red) at single or both chromatids. Pie chart representing % of chromosomes having EdU signal at telomeres at a single chromatid or both chromatids. Source data are provided as a Source Data File.

The online version of this article includes the following source data for figure 5:

**Source data 1.** Quantification of BrdU-TRF2 co-localisation in S and non-S nuclei.
**Source data 2.** Quantification of Mitosis DNA synthesis in *TRF1^{F/F}* MEFs and at telomeres.

cell lines. Loss of SMC5 and TRF1 expression were confirmed by immunoblotting (*Figure 6A–B*), while mRNA levels of POLD3 were analysed by RT-QPCR (*Figure 6C*). We only noticed a slight decrease in population doublings in the double mutants, while all cell lines were still able to properly divide and incorporate EdU (*Figure 6—figure supplement 1, A–B*). Thus, we carried out EdU-FISH in these cells to check for the presence of BIR (*Figure 6D*). We found that the enrichment of DNA synthesis at telomeres in *TRF1* deleted cells was suppressed in the double mutant *TRF1-POLD3*, while the double mutant *TRF1-SMC5* revealed similar telomeric DNA synthesis when compared to the single *TRF1* mutant (*Figure 6E*). First, these results confirm that BIR is the molecular mechanism taking place at TRF1 depleted telomeres. Second, SMC5 appears to be dispensable for BIR dependent DNA synthesis at these replication-stressed chromosome ends.

## SMC5 and POLD3 are required for APBs formation and recombination at TRF1 deficient telomeres

We further examined whether POLD3 and SMC5 could be responsible not only for the BIR dependent DNA synthesis but also for the other ALT-like phenotypes observed at TRF1 deficient telomeres.

Since TRF1 is well known to suppress telomere fragility or MTS (*Sfeir et al., 2009*) (*Martínez et al., 2009*), we first investigated the role of POLD3 and SMC5 in the induction or maintenance of this telomere replication stress in the double mutants (*Figure 6—figure supplement 2, A–B*). As previously reported, TRF1 depleted telomeres present approximately 20% of fragile telomeres per chromosomes (*Figure 6—figure supplement 2, C*). We could not detect any changes in the frequency of telomere fragility in *TRF1-POLD3* nor *TRF1-SMC5* mutants (*Figure 6—figure supplement 2, C*) suggesting that neither POLD3 nor SMC5 are involved in the mechanism that gives rise to telomere fragility. As APBs were increased in *TRF1* deleted cells (*Figure 2C*), we investigated the roles of *POLD3* and *SMC5* in the formation of these specialised bodies. A significant reduction in number of cells having co-localising PML-telomere foci was detected in the double mutant cells *TRF1-POLD3* and *TRF1-SMC5* (*Figure 7A*) suggesting that *POLD3* and *SMC5* are necessary for the formation of these recombination machinery loci. We next explored the involvement of these two factors in HR by scoring for T-SCE (*Figure 7B*), discriminating also between the two categories of T-SCEs (single or double exchanges) in the analysis of the double mutants *TRF1-SMC5* and *TRF1-POLD3*. We found that both types of exchanges are dependent on SMC5 and POLD3 (*Figure 7—figure supplement 1*). Finally, we assessed TERRA expression levels in the double mutants. Surprisingly, only the absence of *POLD3* was able to rescue the increase in TERRA levels detected in *TRF1* deficient cells, while the *SMC5* single mutant increased TERRA expression (*Figure 7C*). Collectively, our data indicate that both POLD3 and SMC5 are essential for T-SCE and APBs formation, but only POLD3 is required to maintain increased TERRA levels and BIR observed in *TRF1* deficient cells. This suggests that POLD3 and SMC5 have separate roles or act at different stages of the recombination events happening at TRF1 depleted telomeres, advocating also an intriguing connection between

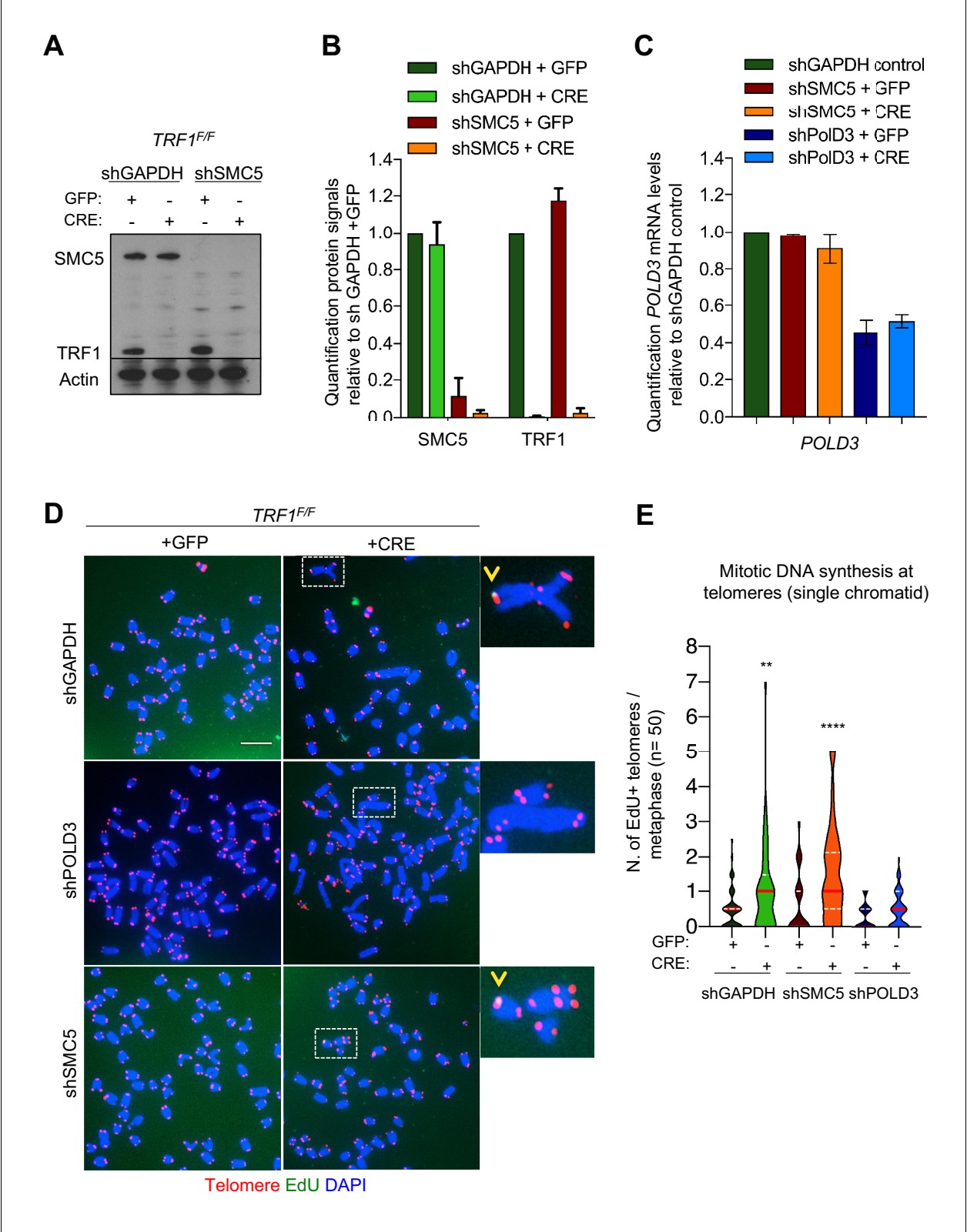

**Figure 6.** POLD3 but not SMC5 regulates mitotic DNA synthesis at *TRF1* deleted telomeres. (**A**) Western blotting showing expression of SMC5, TRF1 and Actin (loading control) proteins in *TRF1^F/F^* MEFs after infection with GFP or CRE-Adenovirus and deletion of *SMC5* by shRNA. shGAPDH is used as negative control. (**B**) Quantification of the knock-out and knock-down shown in A. Graph shows protein signal quantification relative to shGAPDH in +GFP control cells, data are represented as mean ± SEM of 3 independent biological replicates. (**C**) Quantification of *POLD3* mRNA levels relative to
*Figure 6 continued on next page*

*Figure 6 continued*

*GAPDH* control. Data are represented as mean ± SEM of 3 independent biological replicates. (**D**) Representative images of 6 different genotypes generated in the above description. Metaphases show EdU (green), telomeres labelled with TelPNA-C-rich-Cy3 (red) and chromosomes counterstained with DAPI (blue). Scale bar, 10 µm. (**E**) Quantification of mitotic DNA synthesis at telomeres (single chromatid) in *TRF1^F/F^* MEFs infected with shGAPDH control (GFP or CRE), shSMC5 (GFP or CRE) and shPOLD3 (GFP or CRE). Data are represented (n = 50 metaphases) as number of EdU positive telomeres per metaphase with a violin plot, where the median is underlined in red and quartiles in white. One-way ANOVA multiple comparisons (**, p<0.01; ****, p<0.0001) relative to shGAPDH+GFP sample. Source data are provided as a Source Data File.

The online version of this article includes the following source data and figure supplement(s) for figure 6:

**Source data 1.** WB and Quantification of SMC5 knock-down efficiency in *TRF1^F/F^* MEFs.
**Source data 2.** POLD3 mRNA levels after KD in *TRF1^F/F^* MEFs.
**Source data 3.** Quantification of Mitosis DNA synthesis at telomeres in *TRF1^F/F^* MEFs with and without POLD3 and SMC5.
**Figure supplement 1.** Cell proliferation and EdU incorporation are not affected in *TRF1, TRF1-SMC5* and *TRF1-POLD3* mutants.
**Figure supplement 1—source data 1.** Population doublings in *TRF1^F/F^* MEFs with and without POLD3 and SMC5.
**Figure supplement 1—source data 2.** Number of S- and non-S phase in *TRF1^F/F^* MEFs with and without POLD3 and SMC5.
**Figure supplement 2.** SMC5 and POLD3 are dispensable for TRF1 dependent telomere fragility.
**Figure supplement 2—source data 1.** Fragile telomeres in *TRF1^F/F^* MEFs with and without POLD3 and SMC5.

TERRA and BIR. We speculate that TERRA could trigger the homology search by stimulating the initial steps of BIR in which POLD3 is involved (*Figure 8*).

## Discussion

Faithful DNA replication of genetic information is essential for the maintenance of genome stability and integrity. Specific genomic loci, including fragile sites and telomeres, represent major obstacles to DNA replication progression and/or completion. Fragile sites have the propensity to form visible gaps or breaks on chromosome in metaphase spreads of cell lines from patients having fragile X-syndrome or Huntington's disease (reviewed in *Minocherhomji and Hickson, 2014*; *Minocherhomji et al., 2015*). It is well documented that CFS expression is exacerbated in cells grown under low to mild replication stress, for example upon inhibition of DNA polymerase with APH (*Minocherhomji and Hickson, 2014*; *Minocherhomji et al., 2015*). Fragile sites are hotspots for deletions, chromosome rearrangements and are associated with an increased frequency of homologous recombination (*Glover and Stein, 1987*). Over the last decade, telomeres have been identified as APH induced fragile sites displaying the standard phenotype of multiple spatially distinct telomere foci (MTS or telomere fragility) on metaphase spreads (*Martínez et al., 2009*; *Sfeir et al., 2009*). Various factors suppress MTS and thereby facilitate DNA replication at telomeres; including (*Zaaijer et al., 2016*) the shelterin protein TRF1 (*Martínez et al., 2009*; *Sfeir et al., 2009*), the DNA helicases RTEL1 (*Vannier et al., 2012*), BLM and WRN (*Barefield and Karlseder, 2012*), Topoisomerase TopoIIα (*d'Alcontres et al., 2014*) and Rif1 (*Zaaijer et al., 2016*). High levels of DNA damage and telomere fragility are characteristics of ALT cells (*Cesare et al., 2009*) (*Min et al., 2017*), presenting several DNA repair and damage factors in APBs (*Draskovic et al., 2009*; *Wu et al., 2000*; *Yeager et al., 1999*), indication of elevated telomeric stress in these cells. Therefore, it has been hypothesized that ALT mechanism arises from persistent replication stress, which can be resolved by the initial collapse of the replication fork, subsequently offering substrates for HR repair mechanisms dependent on homology search and telomere synthesis as reported with BIR pathway (*Dilley et al., 2016*).

In this study, we report that replication stress generated at TRF1 depleted telomeres in telomerase positive MEFs is associated with the recruitment of ALT signature factors including PML, subunits of the NuRD complex, BRCA1 and SMC5/6 complex. We suggest that the formation of permissive telomeric chromatin enables transcription of telomeric sequences into TERRAs and increases recombination as measured by T-SCEs, in a POLD3 and SMC5/POLD3 dependent manner, respectively. Moreover, we detect mitotic DNA synthesis at TRF1 depleted telomeres, which is dependent on POLD3 but not SMC5. Comparison analysis with APH-treated MEFs highlighted the unappreciated role of mouse TRF1 in suppressing telomere chromatin re-organisation, which we propose to contribute to HR-dependent telomere-sister chromatid exchanges. Herein, we showed that telomere replication stress induced by TRF1 deletion generates additional telomere responses

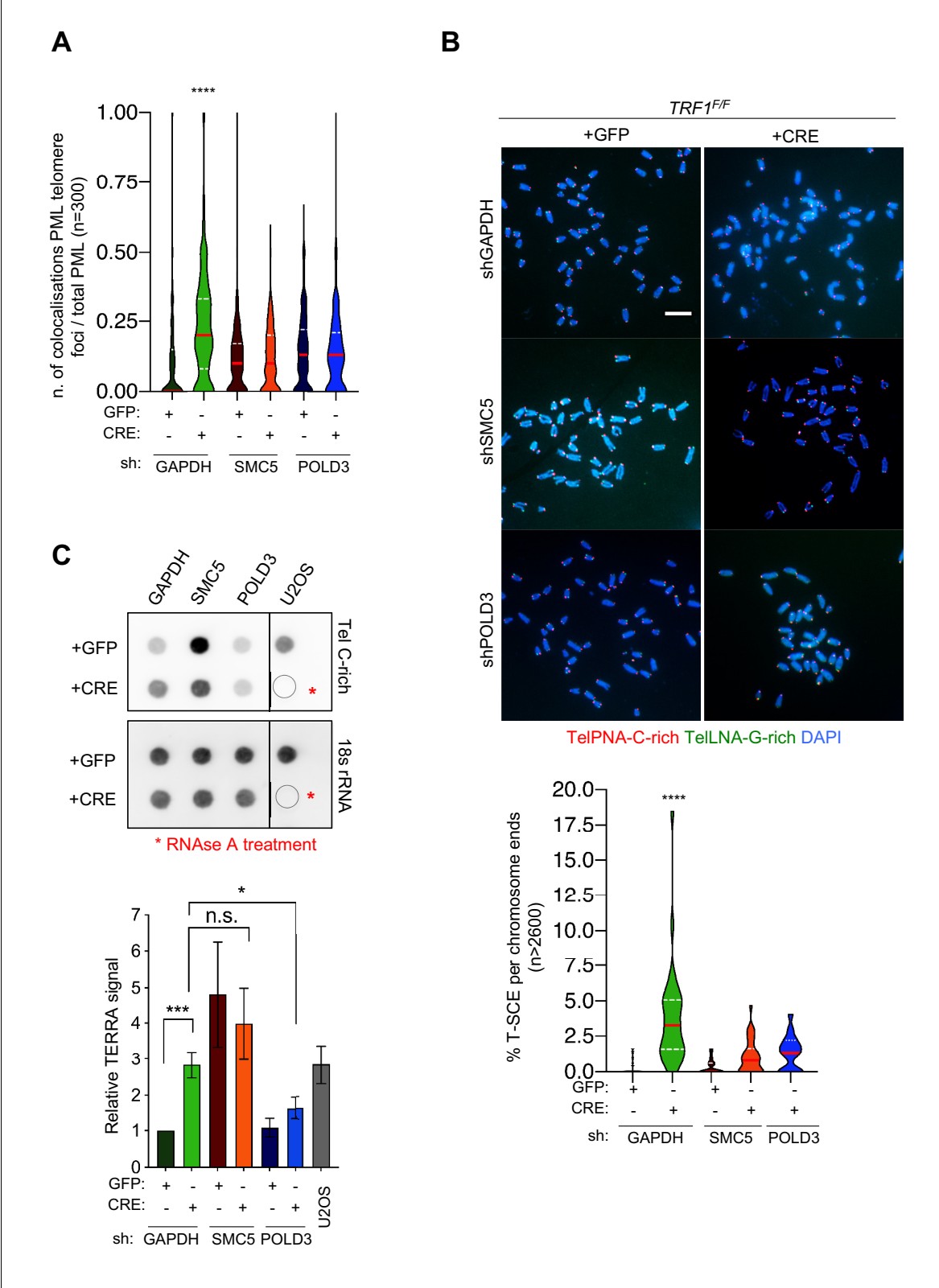

**Figure 7.** *SMC5* and *POLD3* are required for induction of recombination at *TRF1* deficient telomeres. (A) APBs formation in *TRF1* deleted cells is rescued in double mutants *TRF1-SMC5* and *TRF1-POLD3*. Quantification of APBs formation is represented as number of co-localising PML-telomere foci divided by the total number of PML present per nucleus (n = 300 nuclei analysed) from three independent biological replicates. Data are represented with a violin plot, where the median is underlined in red and quartiles in white. One-way ANOVA multiple comparisons (****, p<0.0001)

*Figure 7 continued on next page*

Figure 7 continued

relative to shGAPDH+GFP sample. (B) Representative images of the chromosome oriented (CO)-FISH assay with denaturation, used to score for telomeric T-SCEs in *TRF1^F/F* MEFs infected with shGAPHH control (GFP or CRE), shSMC5 (GFP or CRE) and shPOLD3 (GFP or CRE). Telomeres are labelled with TelPNA-C-rich-Cy3 (red) and TelLNA-G-rich-FAM (green), while chromosomes are counterstained with DAPI (blue). Scale bar, 10 μm. For quantification T-SCE was considered positive when involved in a reciprocal exchange of telomere signal with its sister chromatid (both telomeres yellow) and for asymmetrical exchanges at single chromatid (one telomere yellow). Data are indicated as % of T-SCE per sister telomere (bottom panel). Data (n = > 2600 chromosome ends) from three independent biological replicates are indicated in a violin plot, where the median is underlined in red and quartiles in white. One-way ANOVA multiple comparisons (****, p<0.0001) relative to shGAPDH+GFP sample. (C) RNA dot blot analysis in *TRF1*, *SMC5, POLD3* single and double mutants. The blot was revealed with a DIG-Tel-C-rich probe or 18 s rRNA as a control. TERRA signals were normalised to 18 s rRNA and GFP control (bottom panel). Data are represented as relative TERRA signal ± SEM of 4 independent biological replicates. P values, two-tailed student t-test (*, p<0.05; ***, p<0.001; n.s. = non significant). Source data are provided as a Source Data File.

The online version of this article includes the following source data and figure supplement(s) for figure 7:

Source data 1. APBs co-localisations quantification in *TRF1^F/F* MEFs with and without POLD3 and SMC5.
Source data 2. T-SCEs quantification in *TRF1^F/F* MEFs with and without POLD3 and SMC5.
Source data 3. Quantification of Telomeric RNA molecules by dot-blot in *TRF1^F/F* MEFs with and without POLD3 and SMC5.
Figure supplement 1. SMC5 and POLD3 are required for TRF1 dependent telomere recombination.
Figure supplement 1—source data 1. T-SCEs quantification in *TRF1^F/F* MEFs with and without POLD3 and SMC5.

(APBs, Chromatin remodeling, HR) compared to APH treated MEFs. We propose this is imputed to the function of TRF1 in stabilising telomere chromatin. Both mouse and human TRF1 suppresses telomere fragility and therefore is essential to facilitate DNA replication through telomeric repeats (*Lee et al., 2018*; *Sfeir et al., 2009*). We now show that like for mouse TRF1, its human homolog is essential to suppress APBs formation and BIR, that is POLD3 dependent in mouse and likely to contribute to the repair of stalled/collapsed telomeric forks.

Mouse TRF1 is essential for cell proliferation and to counteract TERRAs expression/stability and activation of HR, while it seems dispensable in HT1080-ST and HeLa cells. This could be explained by the difference in chromatin composition between the two species, indeed human telomeres display less abundant repressive marks compared to mouse chromosome ends (*García-Cao et al., 2004*; *O'Sullivan et al., 2010*; *Rosenfeld et al., 2009*). Moreover, mouse and human TERRAs have different origins. The main source of human TERRA molecules is from Human 20q subtelomere (*Montero et al., 2016*) however, other subtelomeres might contribute to local regulation and therefore control both *cis* and *trans* regulations (*Tutton et al., 2016*; *Feretzaki et al., 2019*). On the other hand, mouse TERRAs primary origin is more controversial with reports proposing Chr18 subtelomere as the principal source in MEFs (*de Silanes et al., 2014*), while recent literature hints to the subtelomeric regions of ChrX and ChrY (*Chu et al., 2017*). These PAR-TERRA would be more abundant and involved in trans regulation on the other chromosome ends.

Telomere transcription appears to be differently regulated in human and mouse cells and it is tempting to speculate that the difference in chromatin composition between the two species is essential for different transcription regulations. Therefore, it is plausible that human and mouse telomeres bear different accessibility to HR factors due to the distinct heterochromatin properties, that will undoubtedly be an interesting question to address in the future.

We suggest that chromatin remodelling factors such as NuRD-ZNF827 are recruited to mouse TRF1 deficient telomeres to counteract the shelterin instability. This may be explained by analogy with ALT telomeres where telomeric DNA sequence is interspersed with variant repeats (*Conomos et al., 2012*; *Marzec et al., 2015*), which are suggested to cause displacement of shelterin proteins (*Conomos et al., 2013*), thus increasing replication stress and DDR. In this scenario, nuclear receptors bind the interspersed variant repeats and recruit several chromatin remodelling factors including the NuRD complex, which can further alter the telomere architecture by increasing telomere compaction (*Conomos et al., 2014*); perhaps a transient state before stimulating telomere associations and generating more 'open' recombination permissive conditions at telomeres. In line with our findings, repressive chromatin at DSBs has been proposed to facilitate homology search and promote recruitment of HR proteins like BRCA1 (*Khurana et al., 2014*). In addition to BRCA1, we have identified through PICh analysis the SMC5/6 complex specifically recruited at *TRF1* deficient

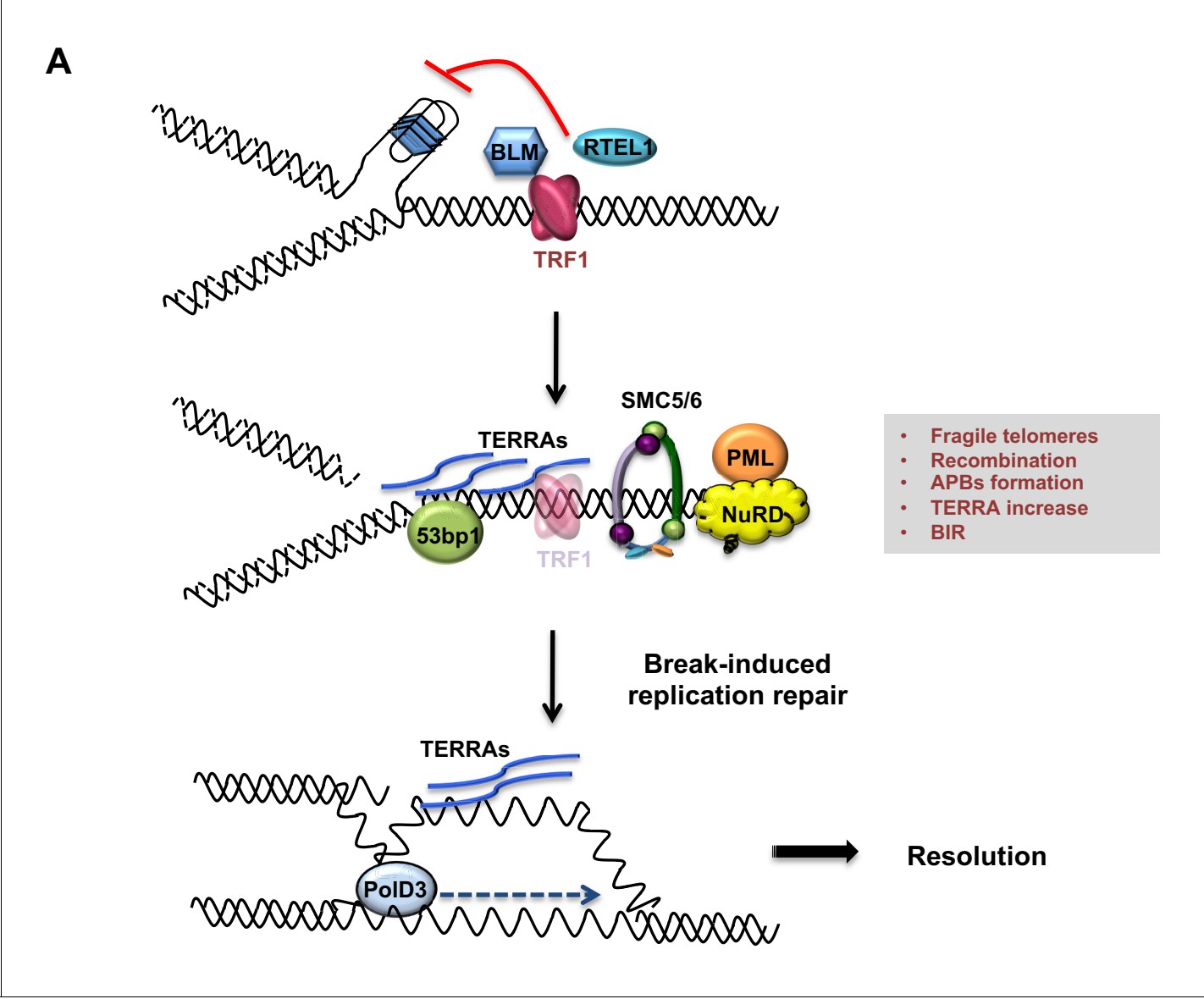

**Figure 8.** Model describing TRF1 as a negative regulator of telomeric transcription (TERRAs), APBs formation, telomeric recombination via PolD3-BIR dependent pathway. Replicative stress induced by TRF1 deletion alters the chromatin status of these telomeres. Recruitment of chromatin remodellers/ HR factors, TERRA accumulation and telomere fragility are observed. The SMC5/6 complex and polymerase POLD3 are among the factors recruited at replicative-stressed telomeres, representing the key players for APBs formation and telomere recombination, particularly BIR-mechanism. We propose that increased TERRAs molecules at telomeres could lead to increased R-loops, which are bypassed by POLD3 dependent BIR to resolve fork progression hindrance.

telomeres. We demonstrate that this complex plays the same role at replication induced telomeres as in ALT cells, targeting telomeres to PML bodies (APBs) and facilitating telomeric HR at these sites (*Potts et al., 2006*), since double mutant *SMC5-TRF1* disrupts formation of APBs and reduces T-SCEs events. However, we were unable to fully induce ALT in *TRF1* deficient MEFs, as they display neither C-circles nor heterogeneity in telomere length and telomerase is still active. The latter could act as a stabiliser of telomeric DNA ends generated during fork restart (*Tong et al., 2015*), similarly to what happens in *RTEL1*[-/-] MEFs (*Margalef et al., 2018*) and in human *RTEL1* deficient cells with long telomeres (*Porreca et al., 2018*).

Persistent DNA damage in ALT cells is suggested to originate from telomeric replication stress, which is proposed to be resolved by BIR in a POLD3 dependent manner (*Dilley et al., 2016*; *Min et al., 2017*; *Roumelioti et al., 2016*). Our results show that TRF1 is a major suppressor of telomeric replication stress and consequently of POLD3 dependent BIR. *TRF1* deficient telomeres present slower movement of S-phase replication forks, measured by molecular combing (*Sfeir et al., 2009*). The slower replication rates at telomeres is proposed to be a consequence of the hindrance of the replication forks by DNA secondary structures, including formation of G-quadruplexes on the lagging strand template or RNA-DNA hybrids. In the absence of TRF1, BLM is unable to be recruited to replicated telomeres and to open DNA secondary structures (*Lee et al., 2018*; *Zimmermann et al., 2014*). Based on our results, we propose that in the absence of TRF1, POLD3 dependent BIR bypasses the stalled replication fork during G2/M phase.

Recent studies identified BIR as a mechanism to bypass RNA-DNA hybrids in a Rad52 and Pol32 dependent manner in yeast (*Amon and Koshland, 2016*; *Neil et al., 2018*). This mechanism is also conserved in human cells where POLD3 is necessary for the restart of stalled replication forks at RNA-DNA hybrids (*Tumini et al., 2016*). Altogether, we propose that increased TERRAs levels at TRF1 depleted telomeres could form RNA-DNA hybrids that are bypassed by POLD3 dependent BIR (*Figure 8*). This is in agreement with recent findings showing that TRF1 suppresses R-loop formation mediated by TRF2 (*Lee et al., 2018*). In contrast to Pold3, SMC5 acts as inhibitor of TERRA accumulation, as its absence is causing a significant increase in TERRA levels. This result is reminiscent of the role of yeast Smc5 in facilitating the resolution of toxic recombination intermediates at RNA-DNA hybrids generated by the helicase Mph1 (*Chen et al., 2009*; *Lafuente-Barquero et al., 2017*). We also describe a role of SMC5 in promoting T-SCEs, but not MiDAS formation (in contrast to PolD3), in the absence of TRF1. These results are indicating an exclusive function of SMC5 in HR at replicative-stressed telomeres, perhaps ensuring the right balance between accumulation and removal of HR-dependent intermediates formed during DNA repair (*Aragón, 2018*). On the other hand, the lack of Smc5 in promoting MiDAS seems in apparent contradiction with a recent observation in ALT cells (*Min et al., 2017*), where telomeric MiDAS is decreased in SMC5/6-depleted Saos2 cells. We speculate, this difference is due to an imbalance of factors used for ALT maintenance, compared to the early events observed in our conditional system after only few population doublings. Therefore, we cannot rule out a possible role of SMC5/6 in promoting MiDAS at a later stage, similar to the one observed in ALT maintenance.

Along with MUS81 structure specific nuclease, POLD3 and POLD4 subunits of the DNA polymerase delta are essential for CFS expression observed in human cells under replication stress (*Minocherhomji et al., 2015*; *Tumini et al., 2016*). To our surprise, *TRF1-POLD3* double mutant did not show any suppression of telomere fragility, indicating key differences in the mechanism generating these phenotypes.

In conclusion, our analysis of TRF1 function provides an unappreciated molecular understanding of the level of protection that this shelterin protein offers at telomeres. Surprisingly, we establish that TRF1 ensures telomeric chromatin stability and suppresses the recruitment of chromatin remodellers including NuRD and the formation of APBs, which together promote a HR prone environment. At the same time, TRF1-dependent replication stress is associated with increased TERRAs expression/stability and activation of POLD3-dependent BIR to restart stalled/collapsed telomeric replication forks.

# Materials and methods

**Key resources table**

| Reagent type (species) or resource | Designation | Source or reference | Identifiers | Additional information |
|---|---|---|---|---|
| Cell line (*Homo-sapiens*) | HT1080-ST fibrosarcoma cell line | kind gift from J Lingner *Cristofari and Lingner, 2006* | | |

*Continued on next page*

*Continued*

| Reagent type (species) or resource | Designation | Source or reference | Identifiers | Additional information |
|---|---|---|---|---|
| Cell line (*Homo-sapiens*) | *TRF1* CRISPR/Cas9 HeLa cell line | kind gift from Z Songyang *Kim et al., 2017* | | Doxycycline-inducible |
| Cell line (*M. musculus*) | *TRF1^{F/F}* Mouse Embryonic Fibroblasts | Established in Boulton Lab (Crick Institute) *Sfeir et al., 2009* | | RRID: CVCL_UE12 Primary and SV40 immortalised |
| Transfected construct (human) - in HT1080-ST | TERF1 siRNA SMARTpool: ON-TARGETplus | Dharmacon/ Thermo Fisher Scientific | #L-010542-02-0020 | 100 nM |
| Transfected construct (human) - in HT1080-ST | ctl siRNA ON-TARGETplus Non-targeting Pool | Dharmacon | #D-001810-10-20 | 100 nM |
| Other | Nucleofector II/2b Device | Lonza | #LO AAB-1001 | programme X-001. |
| Commercial assay, kit | Nucleofector kit T | Lonza | #VVCA-1002 | |
| Transfected construct (Mouse) | pLKO.1-puromycin lentiviral vectors shRNA for mouse SMC5 | Sigma | sequence: CCCATAATGCTCACGATTAAT | in MEFs |
| Transfected construct (Mouse) | pLKO.1-puromycin lentiviral vectors shRNA for mouse POLD3 | Sigma | Sequence: GCATATACTCATGTGTGGTTT | in MEFs |
| Transfected construct (Mouse) | pLKO.1-puromycin lentiviral vectors GAPDH | Open biosystems | Sequence: CTCATTTCCTGGTATGACA | in MEFs |
| Software, algorithm | PRISM eight software, Statistical analysis | GraphPad | | |
| Chemical compound, drug | DAPI stain | Invitrogen | D1306 | (1 µg/mL) |
| Antibody | IgG, Rabbit polyclonal | Abcam | ab37415 | For ChIP 2 µg of antibody |
| Antibody | TRF2, Rabbit polyclonal | Novus | NB110−57130/B2 | For ChIP 5 µg of antibody |
| Antibody | BRCA1, Rabbit polyclonal | Novus | NBP1-45410 | For ChIP 5 µg of antibody |
| Antibody | BAZ1b, Rabbit polyclonal | Cell Signaling | 2152S | For ChIP 5 µg of antibody |
| Antibody | TR4, Mouse monoclonal | R and D biosystems | pp-H0107B-00 | For ChIP 5 µg of antibody |
| Antibody | P66a, Rabbit polyclonal | Novus | NBP1-87359 | For ChIP 5 µg of antibody |
| Antibody | MTA1, Rabbit polyclonal | Abcam | ab71153 | For ChIP 5 µg of antibody |
| Antibody | CHD4, Rabbit polyclonal | Novus | NB100-57521 | For ChIP 5 µg of antibody |
| Antibody | ZNF827, Mouse monoclonal | Santa Cruz | sc514943 | For ChIP 5 µg of antibody |
| Antibody | IgG, Rabbit polyclonal | Abcam | ab37415 | For IF (1:1000) |
| Antibody | TRF2, Rabbit polyclonal | Novus | NB110−57130/B2 | For WB (1:1000) and IF (1:250) |
| Antibody | TRF2, Rabbit polyclonal | Gift from T de Lange | Ref: 1254 | For WB (1:1000) and IF (1:250) |

*Continued on next page*

*Continued*

| Reagent type (species) or resource | Designation | Source or reference | Identifiers | Additional information |
|---|---|---|---|---|
| Antibody | TRF1, Rabbit polyclonal | Gift from T de Lange | Ref: 1449 | For WB (1:1000) and IF (1:250) |
| Antibody | PML, Rabbit polyclonal | Gift from Paul Freemont | / | For IF (1:250) |
| Antibody | SMC5, Rabbit polyclonal | Gift from Jo Murray | / | For WB (1:1000) and IF (1:250) |
| Antibody | Beta-actin, Mouse monoclonal | Abcam | ab8226 | For WB (1:5000) |
| Antibody | BrdU, Mouse monoclonal | MBL | MI-11–3 | For IF (1:500) |
| Antibody | Anti-rabbit Alexa 488 antibody, Donkey polyclonal | Thermo, | A21206 | For IF (1:2000) |
| Antibody | Anti-mouse Ig-HRP, Goat polyclonal | DAKO | P0447 | For WB (1:5000) |
| Antibody | Anti-Rabbit Ig-HRP, Pig polyclonal | DAKO | P0217 | For WB (1:10000) |

For all quantification of telomeric phenotypes, masking was used during data analysis.

## Cell culture, viral transductions and transfections with siRNAs

TRF1 conditional knock-out MEFs (SV40-immortalised) were described previously (*Martínez et al., 2009*; *Sfeir et al., 2009*). Cells were cultured at 37°C in 5%CO2, using DMEM medium supplemented with 10% FCS (Sigma F2442). To achieve TRF1 deletion, cells were infected twice at 72 hr interval with Ad5-CMV-CRE (m.o.i. of 50) and harvested 3 or 4 days after the second infection.

pLKO.1-puromycin lentiviral vectors containing shRNAs for SMC5 (sequence CCCATAATGC TCACGATTAAT, Sigma), POLD3 (GCATATACTCATGTGTGGTTT, Dharmacon) or GAPDH (CTCA TTTCCTGGTATGACA, Open biosystems) were introduced by infection of lentivirus-containing supernatant from 293FT cells. Puromycin selection was performed for 3 weeks at 2 µg/ml and several clones were expanded and cultured before screening them for knock-down efficiency.

HT1080-ST human fibrosarcoma cell line was previously described in *Cristofari and Lingner (2006)* and was a kind gift from J. Lingner. siRNAs transfections were performed in HT1080-ST cells using Cell Line Nucleofector kit T (Lonza #VVCA-1002) according to the manufacturer's instructions. Briefly, 2 million HT1080-ST cells were transfected using 100 nM of SMARTpool: ON-TARGETplus TERF1 siRNA (Dharmacon #L-010542-02-0020) or ON-TARGETplus Non-targeting Pool (Dharmacon #D-001810-10-20). Transfection experiments were carried out using Nucleofector II/2b Device (Lonza #LO AAB-1001), programme X-001. Cells were harvested, counted and transfected again every 48 hr until final collection at day 6.

Doxycycline-inducible TRF1 CRISPR/Cas9 HeLa cell line was a kind gift from Z. Songyang and were previously described in *Kim et al. (2017)*. Cells were incubated in 1 µg/ml of doxycycline for 15 days and the media containing fresh doxycycline was replaced every 2 days.

HeLa (Dox inducible TRF1 KO-CRISPR/Cas9) from Zhou Songyang (*Kim et al., 2017*); mycoplasma contamination testing status: negative. MEFs (*TRF1^{F/F}*) established by Vannier at London Research Institute (now Crick Institute-Boulton Lab) and with cell services, mycoplasma contamination testing status: negative. HT1080 (originally *Mao et al., 1993*) is a kind gift from Joachim Lingner, mycoplasma contamination testing status: negative.

## Aphidicolin treatment

MEFs were treated with 0.4 µM of Aphidicolin (Sigma A0781) for 72 hr before collection and RNA dot blot analysis. For CO-FISH experiments, cells were treated with Aphidicolin for 56 hr before addition of fresh media containing BrdU/C and metaphase collection was performed after 16 hr. In

all the experiments performed using Aphidicolin, media containing fresh drug was changed every 24 hr.

## Western blot

Cells were scraped in cold PBS, spun down and incubated in lysis buffer (NaCl 40 mM; Tris 25 mM, pH 8; MgCl 2 mM; SDS 0.05%; Benzonase 1 µl/2 ml; Complete protease inhibitor cocktail, EDTA-free, Roche) for 10 min on ice. The lysates were sheared 10 times by forcing it through a 25G needle and left on ice for another 10 min. 35 µg of protein lysates were denatured for 10 min at 95°C after addition of Laemmli buffer 4X (50 mM Tris pH7; 100 mM DTT; 2%SDS; 0.1%bromophenol blue; 10% glycerol), separated on 4–12% Bis-Tris gels (Invitrogen) and transferred onto a nitrocellulose membrane (Amersham Protran 0.2 µm NC). Rabbit anti-TRF1 (gift from Titia de Lange) and rabbit anti-SMC5 (gift from Jo Murray) antibodies were diluted in PBST (PBS1x; 0.1% Tween-20, Sigma-Aldrich) with 5% non-fat milk. For detection of human TRF1, rabbit anti-TRF1 (SantaCruz #sc-6165-R) antibody was used at a final concentration of 0.2 µg/ml diluted in 5% non-fat milk (w/v) in TBS-T. Following incubations with HRP-coupled secondary antibodies signals were visualised using ECL II kit (Pierce) and x-ray film exposure (Amersham Hyperfilm ECL). Beta-actin antibody was used for normalisation (Abcam, ab8226).

## Quantitative RT-PCR

RNA extraction was carried out using RNeasy Mini Kit (Qiagen). 500 ng of RNA were subjected to reverse transcription using random hexamer primers and cDNA Synthesis Kit (Roche) according to the manufacturer's protocol. Quantitative PCR was performed using QuantiTect SYBR Green PCR Master Mix and the following primers: mouse POLD3 with antisense 5'-ACACCAAGTAGGTAACA TGCAG-3' and sense 5'-AAGATCGTGACTTACAAGTGGC-3' sequences; Mouse Actin with antisense 5'-CCAGTTGGTAACAATGCCATGT-3' and sense 5'-GGCTGTATTCCCCTCCATCG-3' sequences; The PCR cycles were as follows: 95°C for 15 min, 95°C for 15 s, 55°C for 30 s, 72°C for 30 s for 44 cycles.

## Telomeric chromatin isolation by PICh

PICh was carried out as previously described (*Déjardin and Kingston, 2009*) using the following 2'Fluoro-RNA probes for hybridisation: Destiobiotin-108 atom tether-UUAGGGUUAGGGUUAGGG UUAGGGt (Telo probe); Destiobiotin-108 atom tether-GAUGUGGAUGUGGAUGUGGAUGUGg (Scramble probe).

## Gel and post digestion processing

Gels were processed using a variant of the in-gel digestion procedure as described in *Shevchenko et al. (2006)*. Briefly, gel sections were excised and chopped into uniform cubes, followed by de-staining with 50/50, 50 mM ammonium bicarbonate (AmBic)/acetonitrile(ACN). Gel sections were then dehydrated with 100% ACN followed by the subsequent sequential steps: reduction with 10 mM dithiothreitol (DTT) at 56°C for 30 min in the dark, dehydration, alkylation with 55 mM iodoacetamide (IAM) at RT for 20 min in the dark and dehydration. Gel sections were finally rehydrated with a 40 mM AmBic, 10% ACN solution containing 500 ng of Trypsin Gold (Promega, V5280) and incubated overnight at 37°C.Recovered gel digest extracts were dried on a speed-vac, reconstituted with 99/1, H2O/ACN + 0.1% FA and de-salted using a standard stage tip procedure using C18 spin tips (Glygen Corp, TT2C18). Dried gel digest peptide extracts solubilised in 25 µl of 0.1% trifluoroacetic acid (TFA) and clarified solution transferred to auto sampler vials for LC-MS analysis.

## Mass spectrometry analysis

Peptides were separated using an Ultimate 3000 RSLC nano liquid chromatography system (Thermo Scientific) coupled to a LTQ Velos Orbitrap mass spectrometer (Thermo Scientific) via an EASY-Spray source. 6 µL of sample was loaded in technical duplicates onto a trap column (Acclaim PepMap 100 C18, 100 µm × 2 cm) at 8 µL/min in 2% acetonitrile, 0.1% TFA. Peptides were then eluted on-line to an analytical column (EASY-Spray PepMap C18, 75 µm × 25 cm). Peptides were separated using a linear 120 min gradient, 4–45% of buffer B (composition of buffer B– 80% acetonitrile, 0.1% formic

acid). Eluted peptides were analysed by the LTQ Velos operating in positive polarity using a data-dependent acquisition mode. Ions for fragmentation were determined from an initial MS1 survey scan at 15000 resolution (at m/z 200), followed by Ion Trap CID (collisional induced dissociation) of the top 10 most abundant ions. MS1 and MS2 scan AGC targets set to 1e6 and 1e4 for a maximum injection time of 500 ms and 100 ms respectively. A survey scan m/z range of 350–1500 was used, with a normalised collision energy set to 35%, charge state rejection enabled for +one ions and a minimum threshold for triggering fragmentation of 500 counts.

## Data analysis

All data files acquired were loaded into MaxQuant (*Cox et al., 2014*) version 1.6.0.13 analysis software. Raw files were combined into an appropriate experimental design to reflect technical and biological replicates. The LFQ algorithm and match between runs settings were selected. Data were searched against the UniProt Reference Proteome *Mus musculus* protein database (UP000000589), downloaded on 16th January 2019 from the UniProt website. The database contains 17,002 reviewed (Swiss-Prot) and 37,186 un-reviewed (TrEMBL) protein sequences. MaxQuant also searched the same database with reversed sequences so as to enable a 1% false discovery rate at peptide and protein levels. A built-in database of common protein contaminants was also searched.

Upon completion of the search, the 'proteingroups.txt' output file was loaded in Perseus version 1.4.0.2. Contaminant and reverse protein hits were removed. LFQ intensities were log2 transformed. Data were group categorised to 'Scramble', 'Telomere' or 'Deletion'. Data were filtered for a minimum of 3 valid LFQ intensity values in at least one group. Missing values (NaN) were imputed from a normal distribution with default values.

## FISH and CO-FISH on metaphase spreads

For metaphase spread preparation, cells were incubated for 60 min with 10 ng/ml colcemid (Roche). Cells were harvested, swollen in 75 mM KCl solution for 15 min at 37°C, fixed in ethanol/acetic acid solution (3:1, v/v) and washed three times with the same fixing solution. Suspensions of fixed cells were dropped onto glass slides and dried overnight before performing FISH experiments.

Q-FISH and CO-FISH procedures were performed as previously described (*Ourliac-Garnier and Londoño-Vallejo, 2011*). Briefly, metaphase spreads were fixed in 4% formaldehyde for 2 min, washed 3 × 5 min in PBS 1x, treated with pepsin (1 mg/ml in 0.05 M citric acid pH 2) for 10 min at 37°C, post-fixed for 2 min, washed and incubated with ethanol series (70%, 80%, 90%, 100%). Hybridising solution containing Cy3-O-O-(CCCTAA)$_3$ probe (PNA bio) in 70% formamide, 10 mM Tris pH 7.4% and 1% blocking reagent (Roche, 11096176001) was applied to each slide, followed by denaturation for 3 min at 80°C on heating block. After 2 hr hybridisation at RT, slides were washed twice 15 min in 70% formamide, 20 mM Tris pH 7.4, followed by three washes of 5 min in 50 mM Tris pH 7.4, 150 mM NaCl, 0.05% Tween-20, dehydrated in successive ethanol baths and air-dried. Slides were mounted in antifade reagent (ProLong Gold, Invitrogen) containing DAPI and images were captured with Zeiss microscope using Carl Zeiss software. Telomeric signals were quantified using the ImageJ software (Fiji).

For CO-FISH, the cells were treated with 10 µM BrdU:BrdC (3:1) for 16 hr, followed by colcemid treatment as above. Prior to hybridisation slides were treated with RNAse A (0.5 µg/ml in PBS) for 10 min at 37°C, incubated with Hoechst (1 µg/ml in 2XSSC) for 10 min at RT, exposed to UV light for 1 hr and treated with *Exo*III to degrade the neosynthesised DNA strand containing BrdU/C. Slides were next dehydrated through ethanol series, hybridising solution containing TelG-FAM probe (Exiqon) in 50% formamide, 2XSSC, 1% blocking reagent was applied to each slide, followed by denaturation for 3 min at 80°C on heating block and hybridisation for 2 hr in the dark. Slides were washed 2 × 15 min in 50% formamide, 2XSSC and 3 × 5 min in 50 mM Tris pH 7.4, 150 mM NaCl, 0.05% Tween-20. Finally, slides were dehydrated, incubated with TelC-cy3 probe for 2 hr, followed by the steps described above in the FISH protocol.

## Immunofluorescence-FISH

Cells seeded on slides were permeabilised with Triton X-100 buffer (0.5% Triton X-100; 20 mM Tris pH8; 50 mM NaCl; 3 mM MgCl2; 300 mM sucrose) at RT for 5 min and then fixed in 3% formaldehyde/2%sucrose in PBS1X for 15 min at RT and washed three times in PBS1X. After a 10 min

permeabilisation step and a wash in PBS1X, nuclei were incubated with blocking solution (10% serum in PBS1X) for 30 min at 37°C and stained with specific primary antibodies: rabbit anti-PML (1/200, a gift from Paul Freemont); rabbit anti-53bp1 dilution (1/400, Bethyl A300-272A). After three washes in PBS1X, nuclei were incubated with secondary donkey anti-rabbit Alexa 488 antibody (1/400, Life Technologies) for 40 min at 37°C, washed three times in PBS1X, post fixed 10 min and hybridised with TelC-cy3 PNA probe as described in FISH protocol.

EdU labeling and staining were performed as previously reported (*Minocherhomji et al., 2015*). Briefly, cells were incubated 1 hr with EdU (100 µM) and colcemid (10 ng/ml), followed by metaphase spread preparation. For EdU staining, the steps of fixation, pepsin treatment and dehydration in ethanol serial dilutions were carried out as in FISH protocol, followed by Click IT assay using EdU-Alexa Fluor 488 imaging kit according to the manufacturer's instructions (Thermo Fisher). Metaphases were post-fixed and hybridised with TelC-cy3 PNA probe.

## Terra-fish

TERRA-FISH experiment was carried out as previously described (*Azzalin et al., 2007*) with minor modifications. Briefly cells were permeabilised 5 min with cold CSK buffer (10 mM Pipes pH7;100 mM NaCl; 300 mM sucrose; 3 mM MgCl2; 0.5% Triton X-100 and 10 mM of inhibitor Ribonucleoside Vanadyl Complex). After a wash in PBS1X, cells were fixed for 10 min in 3%formaldeyde solution and washed three times with PBS, followed by Immunofluorescence with primary anti-TRF2 (dilution 1/10.000, 1254 ab gift from T. de Lange). Nuclei were then incubated with secondary donkey anti-rabbit Alexa 488 antibody (1/400, Life Technologies) for 40 min at 37°C, washed three times in PBS1X and post fixed for 10 min. After incubation with ethanol series (70%, 80%, 90%, 100%) slides were dried O/N in the dark. TelC-cy3 PNA probe was used for TERRA detection and after incubation for 2 hr at RT, slides were washed 3 × 5 min in 50% formamide, 2XSSC at 39°C, 3 × 5 min in 2XSSC at 39°C and a final wash in 2XSSC at RT. Slides were dehydrated in successive ethanol baths, air-dried and mounted in antifade reagent (ProLong Gold, Invitrogen) containing DAPI and images were captured with Zeiss microscope using Carl Zeiss software. Quantification was performed using CellProfiler 3.1.8 software.

## Chromatin immunoprecipitation (ChIP)

Chromatin preparation and ChIP experiments were performed as previously described (*Porreca et al., 2018*) with the following modifications: sonication of chromatin was performed for 20 min (30 s on/30 s off) in a Diagenode water bath-sonicator at high speed. 20–50 µg of chromatin was diluted 10 times in ChIP dilution buffer (20 mM Tris-HCl pH 8, 150 mM KCl, 2 mM EDTA pH 8, 1% Triton X-100, 0.1% SDS), pre-cleared with Dynabeads (Invitrogen) and incubated overnight with 2–5 µg of antibody (listed in the key resources table).

## RNA dot blot

RNA extraction was carried out using RNeasy Mini Kit (Qiagen), according to the manufacturer instructions. 2 µg of RNA were denatured in 0.2 M NaOH by heating at 65°C for 10 min, incubated 5 min on ice and spotted onto a positively charged Biodyne B nylon membrane (Amersham Hybond, GE Healthcare). Membranes were UV-crosslinked (Stratalinker, 2000 kJ) and baked for 45 min at 80°C, followed by hybridisation at 42°C with digoxigenin (DIG)-labeled telomeric C-rich oligonucleotide TAA(CCCTAA)4, prepared using 3' end labelled kit (Roche). Signal was revealed using the anti-DIG-alkaline phosphatase antibodies (Roche) and CDP-Star (Roche) following the manufacturer's instructions. Images were captured using the Amersham Imager 680 (GE Healthcare) and analysed using the Image Studio Lite software.

18 s rRNA probe with sequence: 5'-CCATCCAATCGGTAGTAGCG was used for normalisation.

## Northern blot

10 µg of RNA was denatured for 10 min at 65°C in sample buffer (50% formamide, 2.2M formaldehyde, 1X MOPS) followed by ice incubation for 5 min. 10X Dye buffer (50% Glycerol, 0.3% Bromophenol Blue, 4 mg/ml Ethidium Bromide) was added to each sample and all of them were run on a formaldehyde agarose gel (0.8% agarose, 1X MOPS, 6.5% formaldehyde) at 5V per cm in 1X MOPS buffer (0.2M MOPS, 50 mM NaOAc, 10 mM EDTA, RNAse free water). The gel was rinsed twice in

water, washed twice with denaturation solution (1.5M NaCL, 0.05M NaOH), followed by additional three washes with 20XSSC before transferring the RNA on a positively charged Biodyne B nylon membrane (Amersham Hybond, GE Healthcare) using a neutral transfer in 20XSSC. The membrane was fixed and detected as described for the RNA dot blot.

## Acknowledgements

We thank Titia de Lange for providing TRF1 antibody, Paul Freemont for providing PML antibody, Jo Murray for SMC5 antibody, Joachim Lingner for HT1080 cells and Zhou Songyang for TRF1 inducible CRISPR/Cas9 cells. Special thanks to Paulina Marzec for teaching RMP the PICh techniques in Boulton Lab and Jerome Dejardin for the constructive feedback. We thank Arturo Londono, Julia P Cooper, Julian Sale, Titia de Lange, Valerie Borel, Simon Boulton for their helpful comments. TRF1 MEFs were generated in Simon Boulton's group from Titia de Lange's mouse line.

Vannier lab's work is supported by the London Institute of Medical Sciences (LMS), which receives its core funding from UKRI (MRC) and by an ERC Starter Grant (637798; MetDNASecStr). Rosa Maria Porreca is funded by ERC Starter Grant (637798; MetDNASecStr).

## Additional information

### Funding

| Funder | Grant reference number | Author |
|---|---|---|
| European Commission | 637798 MetDNASecStr | Rosa Maria Porreca<br>Emilia Herrera-Moyano<br>Eleni Skourti<br>Pui Pik Law<br>Jean-Baptiste Vannier |
| Medical Research Council | Telomere Replication and Stability | Roser Gonzalez Franco<br>Alex Montoya<br>Peter Faull<br>Holger Kramer<br>Jean-Baptiste Vannier |

The funders had no role in study design, data collection and interpretation, or the decision to submit the work for publication.

### Author contributions

Rosa Maria Porreca, Data curation, Software, Formal analysis, Validation, Investigation, Visualization, Methodology, Help in designing the project, conducted experiments and wrote the manuscript; Emilia Herrera-Moyano, Pui Pik Law, Roser Gonzalez Franco, Data curation, Methodology, Conducted experiments; Eleni Skourti, Formal analysis, Methodology, Conducted experiments; Alex Montoya, Holger Kramer, Data curation, Formal analysis, Methodology, Performed mass spectrometry and its analysis; Peter Faull, Data curation, Formal analysis, Methodology, Performed mass spectrometry and helped with its analysis; Jean-Baptiste Vannier, Conceptualization, Resources, Data curation, Software, Formal analysis, Supervision, Funding acquisition, Validation, Investigation, Visualization, Methodology, Writing - original draft, Project administration, Writing - review and editing, Obtained the grant, Designed the project, Conducted experiments, Analysed the results and wrote the manuscript

### Author ORCIDs

Pui Pik Law  https://orcid.org/0000-0001-8924-0462
Peter Faull  https://orcid.org/0000-0001-8491-8086
Jean-Baptiste Vannier  https://orcid.org/0000-0002-4471-1854

### Decision letter and Author response

Decision letter https://doi.org/10.7554/eLife.49817.sa1
Author response https://doi.org/10.7554/eLife.49817.sa2

## Additional files

### Supplementary files
• Transparent reporting form

### Data availability
All data generated or analysed during this study are included in the manuscript and supporting files. Source data files have been provided for all figures. Proteomic data have been made available at ProteomeXchange, under the accession code PXD017022.

The following dataset was generated:

| Author(s) | Year | Dataset title | Dataset URL | Database and Identifier |
|---|---|---|---|---|
| Porreca RM, Herrera-Moyano E, Skourti E, Law PP, Franco RG, Montoya A, Faull P, Kramer H, Vannier JB | 2020 | Data from: TRF1 averts chromatin remodelling, recombination and replication dependent-Break Induced Replication at mouse telomeres | http://proteomecentral.proteomexchange.org/cgi/GetDataset?ID=PXD017022 | ProteomeXchange, PXD017022 |

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
