## [Decision Letter]

**Acceptance summary:**

Telomeric repeat sequences are prone to induce replication stress and one of the core telomere binding proteins, TRF1, is thought to counteract these problems. The molecular mechanisms coming into play upon TRF1 loss were not clearly defined though. This study combines an inducible TRF1 loss with telomere-targeted proteomics to analyze the incurring changes. The results show that loss of TRF1 causes important alterations in telomere associated proteins and the new profile resembles in a number of ways that of cells in which telomeres are maintained by telomerase-independent mechanisms, named ALT. The presence of DNA repair proteins (MRN complex, the SMC5/6 complex), some increases in telomere colocalization with PML bodies, sister chromatid exchanges and clear signs of increased TERRA transcription and mitotic break-induced replication are part of these phenotypes. The data also show that these early ALT phenotypes can be differentiated from drug induced replication stress, and part of this ALT response is conserved in human cells. These studies therefore point to a critical role for TRF1 in preventing the emergence of ALT and hence immortalization of cells. Given that the details for how ALT actually is initiated remained unclear, these findings are an important step forward and will spur new research into that question.

**Decision letter after peer review:**

Thank you for submitting your article "TRF1 prevents permissive DNA damage response, recombination and Break Induced Replication at telomeres" for consideration by *eLife*. Your article has been reviewed by three peer reviewers, and the evaluation has been overseen by a Reviewing Editor and Jessica Tyler as the Senior Editor.

The reviewers have discussed the reviews with one another and the Reviewing Editor has drafted this decision to help you prepare a revised submission.

Thanks for your submission and support of *eLife*. As mentioned above, your study has been evaluated by three referees and after discussion, they unanimously thought that your manuscript is a nicely crafted proteomics study that does directly support a much speculated, but not evidence based, theory on replication stress causing initial events of the ALT mechanism. The inducible loss of TRF1 is a very efficient way to study these events and the results are viewed as robust and clear. This is an important finding that has the potential to lead to new investigations on how ALT arises and bring our understanding of that to a new level. However, the reviewers also thought that the general relevance of the study would increase substantially, if experiments directed at dissecting the role of TRF1 could be added. Such results would really elevate the study to a prime candidate for *eLife*, where we strive to publish only the most significant and outstanding advances in the field.

Therefore, the reviewers agreed that the following two approaches would be required for the paper to go forward:

1) Are the detected molecular events after TRF1 loss a consequence of chronic replication stress or does TRF1 directly somehow inhibit these events? In order to test these alternatives, one could induce strong replication stress in the presence of TRF1. The experiment would be to treat cells with Aphidicolin (or HU) and assess initial ALT phenotypes directly. SMC5/6 Chip, impact on TERRA, chromatin changes, and ideally some phenotypic outcome at telomeres (SCE exchanges, or mitotic synthesis).

2) Are these events also occurring in human cells suffering replication stress and or loss of TRF1? Reviewers suggest to use a long telomere HeLa variant (1.2.11; or 1.3) and repress TRF1 and/or apply replication stress compounds as above. Readouts again would be characteristic molecular events for ALT.

---

## [Author Response]

Thanks for your submission and support of eLife. As mentiioned above, your study has been evaluated by three referees and after discussion, they unanimously thought that your manuscript is a nicely crafted proteomics study that does directly support a much speculated, but not evidence based, theory on replication stress causing initial events of the ALT mechanism. The inducible loss of TRF1 is a very efficient way to study these events and the results are viewed as robust and clear. This is an important finding that has the potential to lead to new investigations on how ALT arises and bring our understanding of that to a new level. However, the reviewers also thought that the general relevance of the study would increase substantially, if experiments directed at dissecting the role of TRF1 could be added. Such results would really elevate the study to a prime candidate for eLife, where we strive to publish only the most significant and outstanding advances in the field.Therefore, the reviewers agreed that the following two approaches would be required for the paper to go forward:1) Are the detected molecular events after TRF1 loss a consequence of chronic replication stress or does TRF1 directly somehow inhibit these events? In order to test these alternatives, one could induce strong replication stress in the presence of TRF1. The experiment would be to treat cells with Aphidicolin (or HU) and assess initial ALT phenotypes directly. SMC5/6 Chip, impact on TERRA, chromatin changes, and ideally some phenotypic outcome at telomeres (SCE exchanges, or mitotic synthesis).

We thank the reviewers for their constructive suggestions. We treated wild-type MEFs with aphidicolin (APH) in order to induce replication stress, as it is well described to induce telomere fragility or multi-telomeric signal (Sfeir et al., 2009; Martinez et al., 2009). We have now included new paragraphs in the manuscript describing these new experiments.

We treated wt MEFs for 3 days (0.4µM APH) as it was the longest treatment that cells could support. Under these conditions, we could not detect the recruitment of any of the chromatin remodellers and DNA repair factors BRCA1, MTA1, CHD4, ZNF827 and BAZ1b that we characterised at TRF1 depleted telomeres by ChIP-dot blot (Figure 2—figure supplement 1). We could not test for the recruitment of mouse SMC5/6 to telomeres due to the lack of efficient ChIP grade antibodies. However, we looked for all ALT-associated telomere phenotypes present in *TRF1^-/-^*MEFs: APBs formation (Figure 2—figure supplement 1C), TERRAs (Figure 3—figure supplement 1F) and T-SCEs (Figure 2—figure supplement 2B).

1) We observe a general increase of PML foci in wt MEFs treated with APH, however the total number of APBs is not increased in treated cells compared to untreated. On the contrary, the number of PML-telomere co-localisations *vs* total number of PML significantly decreases in treated cells (Figure 2—figure supplement 1C).

2) wt MEFs treated with APH present an increase of TERRA molecules (Figure 3—figure supplement 1, E-F). This is similar to the increase in HMW TERRAs observed in *TRF1^-/-^* MEFs and could be a consequence of the replication stress.

3) Finally, we also quantified T-SCEs in mouse cells treated with APH and identified an increase in single exchanges, which are characteristic of conservative DNA synthesis – Break Induced Replication. However, the absence of reciprocal exchanges between telomere chromatids indicates that APH-dependent replication stress does not trigger an HR activation/response (up to 3 days) and this is specific to the function of TRF1 (Figure 2—figure supplement 2B).

This comparison analysis is essential for the understanding of the direct and indirect molecular events of repair and recombination that are arising at TRF1 depleted telomeres. We propose that in the absence of TRF1, telomeric chromatin re-arrange to a more HR prone environment and this is mediated by Chromatin remodellers including NuRD. Telomeres associated to PML to form APBs, which is essential for HR. At the same time, TRF1-dependent replication stress activates POLD3-dependent BIR to restart stalled/collapsed telomeric replication forks.

2) Are these events also occurring in human cells suffering replication stress and or loss of TRF1? Reviewers suggest to use a long telomere HeLa variant (1.2.11; or 1.3) and repress TRF1 and/or apply replication stress compounds as above. Readouts again would be characteristic molecular events for ALT.

Considering the important information gained by looking at the differences and similarities between APH- and TRF1-dependent replication stress in mouse cells, we then decided to focus our efforts on validating the function of TRF1 in suppressing characteristic molecular ALT events in human cells. We use small RNA interference against TRF1 in HT1080-ST cells (long telomeres) and looked at characteristic molecular events of ALT, as APBs, TERRAs, BIR and HR. We have now included a new paragraph and supplemental figures describing this part (Figure 2—figure supplement 4 and Figure 3—figure supplement 2).

Briefly, we show that:

1) TRF1-depleted HT1080-ST cells (Figure 2—figure supplement 4A) present increased APBs (Figure 2—figure supplement 4B) similar to the mouse cells.

2) We looked for the accumulation of TERRA molecules in HT1080ST (siRNA-day 6) and HeLa (Crispr/Cas9-day 15) cells depleted for TRF1. After 6 days, we could not detect a significant increase in TERRAs (Figure 3—figure supplement 2A) and neither after 15 days culturing HeLa stable TRF1 KO (Figure 3—figure supplement 2B).

3) We measured sister chromatid exchanges at telomeres in HT1080-ST depleted for TRF1 with siRNA. We observed an increase in single exchanges (Figure 2—figure supplement 4C) that we reported to be a mark of BIR in TRF1-deficient murine cells. However, we could not detect any differences for double exchanges between ctl and TRF1 siRNA treated cells (Figure 2—figure supplement 4C).

In conclusion, some of the TRF1 functions are conserved between mouse and human. For both species, TRF1 suppresses telomere fragility and therefore is essential to facilitate DNA replication through telomeric repeats. We now show that like for mouse TRF1, its human homolog is essential to suppress APBs formation and BIR, which is likely to be the activated mechanism to rescue stalled/collapsed telomeric forks.

We have now included a new paragraph in the Discussion section describing the differences between mouse and human TERRAs.